# The genomes of 204 *Vitis vinifera* accessions reveal the origin of European wine grapes

Gabriele Magris[1,2], Irena Jurman[2], Alice Fornasiero [1,2,3], Eleonora Paparelli[1,4], Rachel Schwope[1,2], Fabio Marroni [1,2], Gabriele Di Gaspero [2✉] & Michele Morgante [1,2✉]

In order to elucidate the still controversial processes that originated European wine grapes from its wild progenitor, here we analyse 204 genomes of *Vitis vinifera* and show that all analyses support a single domestication event that occurred in Western Asia and was followed by numerous and pervasive introgressions from European wild populations. This admixture generated the so-called international wine grapes that have diffused from Alpine countries worldwide. Across Europe, marked differences in genomic diversity are observed in local varieties that are traditionally cultivated in different wine producing countries, with Italy and France showing the largest diversity. Three genomic regions of reduced genetic diversity are observed, presumably as a consequence of artificial selection. In the lowest diversity region, two candidate genes that gained berry–specific expression in domesticated varieties may contribute to the change in berry size and morphology that makes the fruit attractive for human consumption and adapted for winemaking.

[1] Department of Agricultural, Food, Environmental and Animal Sciences, University of Udine, Udine, Italy. [2] Istituto di Genomica Applicata, Udine, Italy. [3] Present address: Center for Desert Agriculture, Biological and Environmental Sciences & Engineering Division, King Abdullah University of Science and Technology, Thuwal, Saudi Arabia. [4] Present address: IGA Technology Services, Udine, Italy. ✉email: digaspero@appliedgenomics.org; michele.morgante@uniud.it

Phylogeography of cultivated grapes (*Vitis vinifera* L. subsp. *sativa* (DC.) Hegi, hereafter *sativa*, also known as subsp. *vinifera*[1]) is linked to the history of ancient populations that settled across the Caspian Sea Basin, the Near East and the Mediterranean Basin. Grape cultivation and winemaking began somewhere in the South Caucasus[2], the northern Fertile Crescent, or the Levant[3], following domestication from local forms of the wild ancestor *Vitis vinifera* L. subsp. *sylvestris* (C.C. Gmel.) Hegi (hereafter *sylvestris*) that has a broad geographic distribution consisting of small isolated populations that are scattered across Europe, northern Africa and Western Asia[4,5]. From the cradle of domestication[6], grape cultivars followed a predominant westward pattern of dispersal, driven by human migration and maritime trades, paralleled by a differentiation in use for fresh consumption (table grapes) or winemaking (wine grapes)[7]. Table grapes were also introduced eastwards into Central Asia along land trade routes. The cultivation of domesticated grapes dates back four millennia in the eastern Mediterranean and two millennia in Western Europe, with vegetative propagation becoming more and more prevalent as a mode of reproduction to preserve the genetic identity of valuable accessions that may have arisen from spontaneous crosses. Paleogenomic evidence supports a very early adoption of vegetative propagation and an ancient origin of some of the currently cultivated varieties, with Savagnin Blanc being at least 900 years old[8]. Western European varieties are the foundation of the global wine industry, with ten varieties (Cabernet Sauvignon, Merlot, Tempranillo, Airen, Chardonnay, Syrah, Grenache, Sauvignon Blanc, Pinot Noir, Trebbiano Toscano) accounting for 26% of the vineyards worldwide[9].

Botanical classification of wild and cultivated grapes is all but simple. Wild forms have been classified either as *sylvestris* var. *typica* (spread from the Atlantic coast to the Caucasus, with limited phenotypic differentiation between western and eastern forms, all characterized by small and tomentose leaves) or as *sylvestris* var. *aberrans* that today is rare or almost extinct in Azerbaijan and Turkmenistan but, in the past, was reported to thrive across the southern Caspian Sea Basin and to be distinguished by large and glabrous leaves[4,10]. Cultivated grapevines have been classified in three ecogeographical groups: *orientalis*, *pontica*, and *occidentalis*[10]. *Orientalis* is considered the ancestral group and is supposed to be morphologically more similar to the wild ancestor *sylvestris* var. *aberrans*. *Pontica* is more similar to eastern forms of *sylvestris* var. *typica*, today thriving in South Caucasus, eastern Anatolia and Armenian highlands. The botanist Negrul also proposed an ecogeographical divide between *pontica georgica*, which includes typical Caucasian wine grapes, and *pontica balcanica* that includes representative varieties of the eastern Mediterranean Basin. Hereafter, we refer to *orientalis* and *pontica georgica* as cultivated germplasm of eastern origin. Feral grapes are present in the cradle of domestication as relic populations of transitional forms from *sylvestris* to *sativa*, more similar to the *sylvestris* lineages that underwent domestication[10], hereafter referred to as eastern feral grapes. According to Levadoux[4], feral grapes are occasionally found also in Europe as hybrids between genuinely autochthonous *sylvestris* and escapees from the vineyards (*lambrusques métisses*) or naturalized seedling populations of *sativa* growing outside the agricultural land (*lambrusques coloniales*), hereafter collectively referred to as western feral grapes, that may explain the morphological similarity observed between western *sylvestris* and some local varieties, e.g. varieties of the Lambrusco family[11]. The origin of western European wine grapes is debated[1,12]. According to one hypothesis[1], a reciprocal gene flow between introduced varieties and local wild populations (also called *lambrusques autochtones* according to[4], hereafter referred to as western *sylvestris*) occurred after the westward dispersal of domesticated forms, when the area of grape cultivation overlapped with the European range of distribution of *sylvestris*[1,13]. According to another hypothesis[12], cultivated grapes have at least two origins, one in the East and the other in the West, and western wine grapes may have had their origin in an independent event of domestication from western *sylvestris*.

Exactly which domesticated types were initially introduced for winemaking into the Mediterranean Basin and what has led to the formation of the *occidentalis* wine grapes remain unaddressed questions. To provide answers, we analysed whole-genome sequencing (WGS) data of 204 *V. vinifera* accessions and ten outgroup species, 124 of which were generated in this study. Prior to this study, the analysis of DNA sequence variation in *V. vinifera* has been partially addressed using genotypic data obtained either with SNP chips from hundreds of accessions[1,14–16] or with WGS from a few individuals[17], which allowed the detection of a network of pedigree relationships. WGS of large diversity panels provides several orders of magnitude more information that is not affected by any type of ascertainment bias. Unlike a recent WGS study of Liang and coworkers[18] that focused on *Vitis* species, interspecific hybrids and recently bred *V. vinifera* varieties, here we present a WGS–based analysis of a panel of *V. vinifera* varieties that well represent the cultivated genetic diversity.

In this work, we provide several lines of evidence that support that processes that lead to the creation of the European wine grapes are dominated by introductions of table grapes from the East followed by extensive and frequent gene flow from local wild forms. We also show that genome-wide levels of diversity in cultivated and wild forms are similar, revealing that the domestication bottleneck has had limited effects on the crop germplasm. We identify, in a region of the genome on chromosome 17 that has undergone a selective sweep, a gain-of-function mutation that led to gene expression changes in two genes that may be related to the domestication process. We finally show that current patterns of genetic variation in wine grapes are associated with geographic and climatic variation.

## Results and discussion

**Sequence variation in 204 *V. vinifera* genomes and ten outgroup species.** We resequenced 122 accessions of *V. vinifera* (including *sativa*, *sylvestris* and feral) and two *Vitis* species at a genome coverage ranging from 8-fold to 90-fold (average 26) and obtained archived sequence reads for another 82 *vinifera* accessions and 8 other grape species (Supplementary Data 1 and Supplementary Note 1). Using uniquely mapped paired-end reads, 7,364,288 SNPs were identified in the non-repetitive regions of the cultivated grape genomes (*sativa*, Supplementary Fig. 1a), of which 596,150 were private to single accessions. Forty-eight bona fide wild and 33 feral *vinifera* added 492,256 additional SNPs (Supplementary Note 2). Validation of SNP calls showed low error rates in genotyping for homozygous and heterozygous sites (0.00013% and 0.01019%, respectively (Supplementary Methods 1–2, Supplementary Note 3, and Supplementary Figs. 2–3).

We used a subset of 5,925,766 polymorphic sites that were informative in the outgroup *Muscadinia rotundifolia* as well as in a set of eight American and Asian *Vitis* species to determine the mutation direction, the unfolded site frequency spectrum, and the strength and direction of selective pressure in cultivated varieties by mutation age and mutation type (Supplementary Note 4 and Supplementary Fig. 1b, c). A relatively large proportion of the SNPs (11.9%) that are polymorphic in *vinifera* predate the speciation event that led to the creation of *vinifera* (trans-specific SNPs), suggesting a largely incomplete lineage sorting in the genus *Vitis* (Supplementary Fig. 4). Only 8.2% of the SNPs found

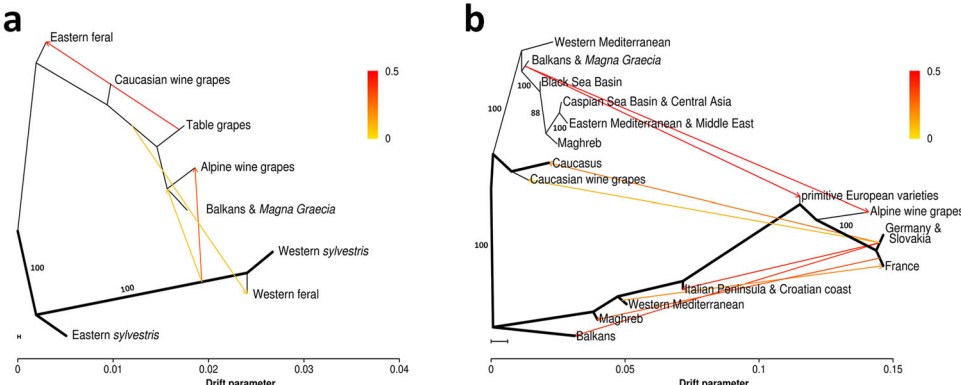

**Fig. 1 Split and admixture events in groups defined by population structure in the WGS panel (a) and by geographic distribution in the diversity panel (b).**
**a** Maximum likelihood (ML) tree with four groups of cultivated varieties (Supplementary Fig. 10) and four groups of wild accessions (Supplementary Fig. 7). Ancestry composition and group sizes are illustrated in Supplementary Fig. 10. **b** ML tree with nine groups of cultivated varieties and seven populations of *sylvestris*. Ancestry composition, group sizes, explained variance and the description of *sylvestris–sylvestris* admixture are given in Supplementary Fig. 22. **a**, **b** Migration events are indicated by colored arrows. The color scale shows the migration weight. The scale bar shows ten times the average standard error of the estimated entries in the sample covariance matrix. Bold lines indicate the *sylvestris* branches of the tree. Trees represent random trees and numbers represent bootstrap support values above 70% (100 iterations) before adding migrations. Support for the migration events and the resulting predictive model is given in Supplementary Figs. 20 and 22c, Supplementary Table 1, and Supplementary Data 2.

in *sylvestris* are not present in *sativa*, consistent with a high level of shared variation between wild and cultivated grapes that is expected under a scenario of extensive gene flow and/or with a moderate bottleneck experienced during the domestication process. The cultivated varieties have a nucleotide diversity of $\pi = 7.29 \times 10^{-3}$ and highly heterozygous genomes (Supplementary Fig. 5), with a maximum of 97.1% of total genome length and 96.8% of genes in heterozygous condition in Sauvignon Blanc (Supplementary Fig. 6), despite a mating system dominated by cleistogamy and self-compatibility. The nucleotide diversity in the wild accessions was equal to $3.80 \times 10^{-3}$. While this diversity value may be underestimated due to an incomplete sampling of all the diversity available in *sylvestris*, it still clearly shows that the domestication events that led to the creation of the cultivated varieties, unlike in other fruit crops[19–21], did not lead to significant genome-wide losses of genetic diversity as a consequence of a major genetic bottleneck, confirming and complementing previous estimates based on haplotype diversity[1].

**Population structure and ancestry components.** We used two different data sets and two different approaches to derive inferences on the history, population structure and geographic differentiation in cultivated grapes. We first used a model-based clustering approach[22] (implemented in the software ADMIXTURE) using whole genome sequence data of 203 accessions of *vinifera* (after removing accession KE–06 from this specific analysis, following the classification of this individual as a feral escapee done by Liang and coworkers[18]) to infer their genetic ancestry and a statistical model developed by Pickrell and Pritchard[23] (implemented in the software TreeMix) to infer splits and gene flow between cultivated and wild grapes (Fig. 1a). We then extended the ancestry and gene flow analyses to an additional set of 1241 accessions (hereafter referred to as diversity panel), using a set of 6357 SNPs in common between the whole genome sequenced accessions and the publicly available SNP profiles of the additional accessions (Fig. 1b).

When we applied the model-based clustering to the species germplasm WGS data ($n = 203$, Supplementary Fig. 7), with $K = 2$ we separated eastern and western ancestry. With >0.85 membership proportion, the western ancestry component defined one group that includes exclusively accessions of western

*sylvestris* and western feral grapes. The eastern ancestry component defined the other group that includes eastern wild and feral grapes, Caucasian wine grapes, table grapes, and cultivated varieties from across the Europe's three great southern peninsulas (Iberian, Italian, and Balkan). The rest of the cultivated germplasm, represented by varieties that today are grown in Alpine countries, appears to be the result of admixture. The statistics to estimate the number of ancestral populations suggested that eastern and western ancestry is the main divide, according to the Evanno's test (Supplementary Fig. 8). According to the cross-validation error, $K = 3$ and 4 provide the best predictive model, with $K = 3$ showing only slightly higher cv-error than $K = 4$ (Supplementary Fig. 9). The existence of up to four ancestry components in *V. vinifera* was also considered plausible by Liang and coworkers[18] in a broader context of the genus *Vitis*[18]. With both $K = 3$ and 4, we confirmed the two main components that are dominant in wild grapes, one (yellow, hereafter referred to as W1), which is dominant in western *sylvestris*, and one (blue, hereafter referred to as W2) that is dominant in eastern *sylvestris* (Supplementary Fig. 7). In consideration of the fact that the *aberrans* forms of *sylvestris* should have gone extinct, both these components, with different proportions, should correspond to the *sylvestris typica* ancestry in the East and in the West. Unlike the W1 component, which is found in both eastern and western wine grapes but is predominant only in wild and feral accessions, the W2 component is predominant in Caucasian wine grapes, in table grapes and in European cultivated varieties east of 40°E longitude. With $K = 4$, two additional components (orange and gray) are predicted, with one (orange, hereafter referred to as C1) predominantly found with the W2 component in table grapes, and the other (gray, hereafter referred to as C2) almost exclusively found in wine grapes. These two components, which are detectable in some eastern feral grapes but not in wild grape samples, could be derived from extinct forms of *sylvestris* (i.e., *aberrans*) and deserve special attention to better understand the structure of genetic diversity in the cultivated compartment. The C2 component is detectable in cultivated varieties of the Muscat family as well as in European wine grapes. The C1 component is most frequently associated with table and wine grapes from around the Mediterranean Basin.

**Ancestral populations and botanical groups of cultivated varieties**. In order to test which scenario of ancestral populations is more consistent with the taxonomic treatment of the cultivated compartment, we applied the model-based clustering to the cultivated germplasm alone ($n = 123$, Supplementary Fig. 10). With $K = 2$, we separated one population containing only wine grape varieties from the Alpine countries, which includes 29.3% of all accessions and corresponds to Negrul's *occidentalis*, and one population that consists of 31.7% of all accessions and includes Caucasian wine grapes, table grapes and European varieties from the Southern Balkans and Iberian Peninsula. With $K = 2$, varieties that are typical of the ecogeographical groups *pontica* and *orientalis* clustered in the same ancestral population. $K = 3$ generated one population corresponding to *occidentalis*, including 20.3% of all accessions, and a divide between one population of table grapes and Caucasian wine grapes, including 14.6% of all accessions, and one population that includes varieties from the entire Balkans (including insular Greece), from Southern Italy and from the Iberian Peninsula, representing 13.0% of all accessions. Only $K = 4$ generated a divide between table grapes and Caucasian wine grapes into two ancestral populations that correspond to Negrul's *orientalis* and *pontica georgica*, respectively. The other two ancestral populations with $K = 4$ were represented by wine grapes from the Alpine countries (*occidentalis*) and by varieties from the Balkans, Greece and Southern Italy, largely corresponding to Negrul's *pontica balcanica*. The adoption of $K = 4$ (Supplementary Figs. 10–12) allowed us to obtain groups that reflect, both in terms of number and composition, the divide and the stratification postulated by Negrul (*orientalis*, *pontica georgica*, *pontica balcanica*, *occidentalis*) and widely accepted in grapevine taxonomy.

**Origin of European wine grapes**. TreeMix provided strong evidence for a single eastern origin for the entirety of the cultivated germplasm as well as for an origin of European wine grapes from introgression of western *sylvestris* individuals into the domesticated lineage of *orientalis* grapes (Fig. 1a). The inclusion of this single event of admixture (Supplementary Fig. 13 and Supplementary Note 5) in the model allowed 98.7% of the variance in relatedness among populations to be explained. TreeMix also suggested the occurrence of gene flow in the opposite direction, going from cultivated accessions into wild populations as a consequence of the migration of intermediate forms between wine and table grapes into western wild populations. With all the events of admixture shown in Fig. 1a, which were confirmed by a 3-population test (Supplementary Table 1), the proportion of the variance in the predicted relatedness among populations explained by the model increased to 99.8% and was resilient to different data treatments (Supplementary Figs. 14–20 and Supplementary Note 6). According to TreeMix analysis, Mediterranean wine grapes from the Balkans and *Magna Graecia*, largely corresponding to *pontica balcanica*, appear to be genetically more similar to *orientalis* ancestors (table grapes) than to *pontica georgica* ancestors (Caucasian wine grapes), in partial disagreement with Negrul's hypothesis (Fig. 1a). Principal component analysis, haplotype-based pairwise genetic distance matrices and pedigree networks (see below) lend further support in favor of this statement.

The analysis of the extended set of accessions in the diversity panel provided historical and geographical resolution to this reconstruction. With four ancestry components (Supplementary Fig. 21), we defined eight groups of well differentiated accessions in eight broad geographic areas (Supplementary Fig. 22), excluding varieties with highly admixed ancestry. Caucasian wine grapes were confirmed to be distinct from all other cultivated

varieties and closely related to the local wild accessions. Table grapes as well as wine grape varieties from the Black Sea Basin, the Middle East and the Mediterranean Basin have prevalently W2 and C1 ancestry, with an increase of the C1 component going from east to west (Supplementary Fig. 22). Cultivated varieties across Europe are characterized by the increasing presence of W1 and C2 ancestry components going from south to north (Supplementary Fig. 22). TreeMix analysis and three-population test (Supplementary Data 2) suggested that the presence of W1 ancestry is to be attributed to admixture events between Mediterranean lineages of *sylvestris*, either extinct or not captured by our sample, and introduced varieties most similar to those today grown in the Balkans and *Magna Graecia* (Fig. 1b). We found the highest W1 western *sylvestris* ancestry proportion in old local varieties considered today as autochthonous in the central and northern Italian peninsula, such as Enantio, Lambrusco di Sorbara, Raboso Piave, Fumat, Greco di Tufo, Aglianico, Verduzzo, Welschriesling, and in the widely grown variety Cabernet Franc that is similar to wild forms still present in the Atlantic Pyrenees[24] (Supplementary Fig. 7). We collectively refer to this germplasm as well as to hybrid forms classified in other papers under the designation of *vigne sauvage faux* (false *sylvestris*) as primitive European varieties, which represented the ninth group included in Fig. 1b. Admixture events may have started in Southern Europe as early as in Greek and Roman times. This scenario agrees with previous estimates that western wine grapes and table grapes have diverged for 2.6 K years[17] and with our demographic model that predicts the nadir of effective population size 2 K years before the present (Supplementary Fig. 23) and suggests resumption of sexual reproduction in domesticated grapevines since Roman times. Further admixture may have later involved other *sylvestris* lineages more similar to those found today around the Alpine region (Fig. 1b and Supplementary Data 2).

**Genomic consequences of admixture events**. In order to understand the consequences of post-domestication *sylvestris-sativa* hybridization, we used an ABBA–BABA test[25] to identify genomic regions in wine grapes from the Alpine countries that received introgression from western wild populations before spreading worldwide. We observed widespread rather than localized signals of introgression (Fig. 2a), suggesting that hybridizations occurred multiple times and left pervasive *sylvestris* ancestry across the genome rather than limited to specific loci under adaptive evolution. We vice versa identified some chromosomal regions that appear to be under negative selection against the introgression of wild alleles. These regions could correspond to loci that are particularly important for quality traits. An analysis using $D_A$ distances[26] and phylogenetic trees built separately in 2368 genomic windows across the genome provided additional evidence for widespread effects of introgression, with western *sylvestris* contributions being detected in 37.7% of the genome (Fig. 2b).

Despite the scale of the dispersal and admixture events that have reshaped the continental diversity for millennia, as shown by multiple lines of evidence presented so far, European wine grapes remain connected with table grapes of the Central Asian oases through a highly interconnected network of first-degree or second-degree relationships (Supplementary Fig. 24) that includes all of the 123 cultivated varieties of the WGS panel. We detected 24 parent–offspring and 4 full-sibling relationships, providing conclusive evidence for previously conflicting inferences (Supplementary Figs. 25–29 and Supplementary Note 7). In the diversity panel, 492 varieties spanning the same geographic range were interconnected by 576 parent–offspring relationships

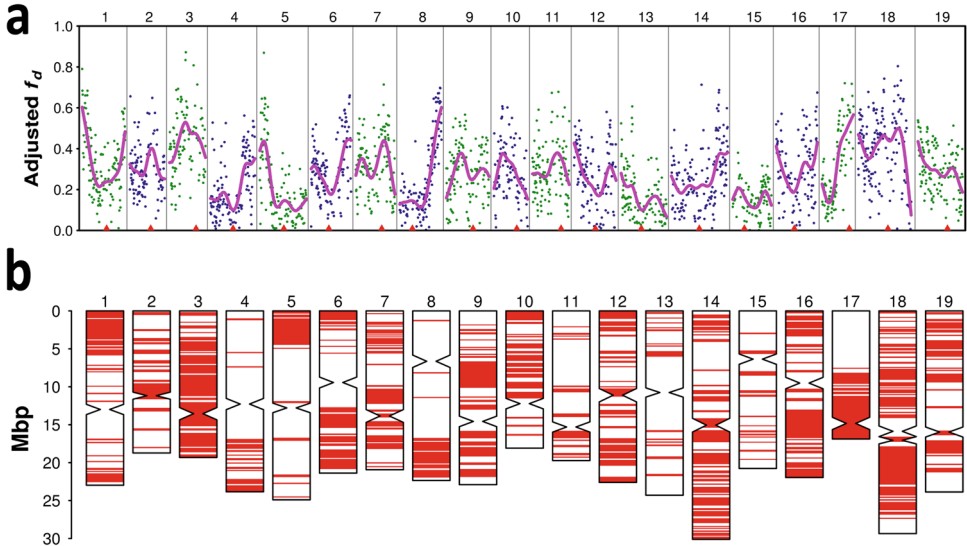

**Fig. 2 Chromosomal patterns of gene flow from western *sylvestris* into table grapes generating Alpine wine grapes. a** Dots represent adjusted $f_d$ values in 100 Kb windows of non-repetitive DNA. Lines represent cubic smoothing splines of the values. **b** Diagram of 100 Kb chromosomal windows (in red) that show phylogenetic tree topologies with shorter genetic distance between Alpine wine grapes and western *sylvestris* than between Alpine wine grapes and any other cultivated group. Red triangles in **a** and constricted regions in **b** indicate the location of centromeric repeats. Source data are provided as a Source Data file.

and another 122 varieties had parent–offspring relationships outside of this network, also including parent–offspring pairs between cultivated varieties and feral grapes (Supplementary Fig. 30).

**Population structure and principal component analysis**. A principal component and coordinate analysis (PCA and PCoA, respectively, in Fig. 3 and Supplementary Fig. 31 and Supplementary Note 8) largely supported the conclusions based on the ancestry analysis about the origin of European wine grapes. The PCA in Fig. 3 shows that the unbiased set of SNPs—obtained by WGS—provided higher resolution for the separation of varieties on the bi-dimensional space than pre-ascertained SNPs used in hybridization-based genotyping. The set of sequenced varieties (Fig. 3a) captured most of the genetic diversity present in the cultivated germplasm as represented in the extended set of accessions (Fig. 3b), including the diversity present in the Iberian peninsula (Supplementary Fig. 32), a center of supposed independent domestication in the West[12]. The PCA in Supplementary Fig. 32 shows that the Iberian cultivated germplasm is more similar to table grapes and Eastern populations of *sylvestris* than to Western populations of *sylvestris*, not providing support to the hypothesized event of neodomestication. These results are consistent with those obtained by Freitas and coworkers[27] from low coverage resequencing of a larger set of locally grown Iberian cultivars and local wild accessions. The PCA highlights individual accessions that may serve as illustrations of the blurred boundaries between wild and cultivated compartments. Enantio and Lambrusco di Sorbara that are cultivated south of the Alps in the Po valley provide an example of western wine grapes that are situated midway on the PCA plane between western *sylvestris* and cultivated varieties from the Balkans and *Magna Graecia* (Fig. 3a) and are contiguous to *sylvestris* accessions from the Italian peninsula (Fig. 3b), as observed by[11]. WGS data show that ADMIXTURE membership proportions in Enantio and Lambrusco di Sorbara (Supplementary Fig. 7) and the level of haplotype sharing with accessions of Western *sylvestris* (Supplementary Fig. 33) are fully compatible with these varieties representing *sylvestris-sativa* first generation hybrids or very early

backcross generations. The feral accession KE–06 from the Ketsch island on the Rhein river (Germany) shows a similar genetic constitution—resulting from a possible cross between an escapee from the vineyards and a genuine autochthonous *sylvestris*, as suggested by[18]—but an opposite case of classification (*lambrusque métis*) presumably because the accession was found outside of a vineyard. There is no evidence of parent–offspring relationships between KE–06 and cultivated varieties of the WGS panel, but we detected the highest level of haplotype sharing with Savagnin Blanc and Pinot Noir (across 40.6 and 39.8% of the diploid genome length, respectively, Supplementary Fig. 34). The Manseng family that was represented in the diversity panel by Gros Manseng, Petit Manseng and Riesling Bleu and is located midway in the PCA plane, as recently observed by[28], between the Pinot/Savagnin Blanc parent–offspring pair and French/German populations of *sylvestris* is in close proximity with an accession classified by Laucou and coworkers[15] as a French *sylvestris* (B00ERBY). The pairs Pinot/B00ERBY, Savagnin Blanc/Petit Manseng, Savagnin Blanc/Riesling Bleu (collection Oberlin), and Petit Manseng/Gros Manseng share a parent–offspring relationship (Supplementary Fig. 30), indicating a possible origin of Petit Manseng, Riesling Bleu and B00ERBY from a cross between a cultivated and a wild accession. Similar hybridization events between cultivated germplasm that was introduced from the center of domestication in the East and local *sylvestris* may also have occurred elsewhere in Southern Europe, generating intermediate forms that somewhere thrive as seedlings in the wild (e.g., feral forms in the Adriatic coast of Croatia, Supplementary Fig. 33) and somewhere are vegetatively propagated for cultivation (e.g., some cultivars in the Iberian peninsula as shown in Supplementary Fig. 32 and proto-varieties in Montenegro[29]). Our analysis of both the WGS panel and the diversity panel did not reveal any instance of cultivated varieties carrying pure Western *sylvestris* ancestry that would be expected in the scenario of an independent domestication event.

**Association between geographic distribution of ancestry components and climate variables**. The history of grape cultivation combines local adaptation with widespread vegetative

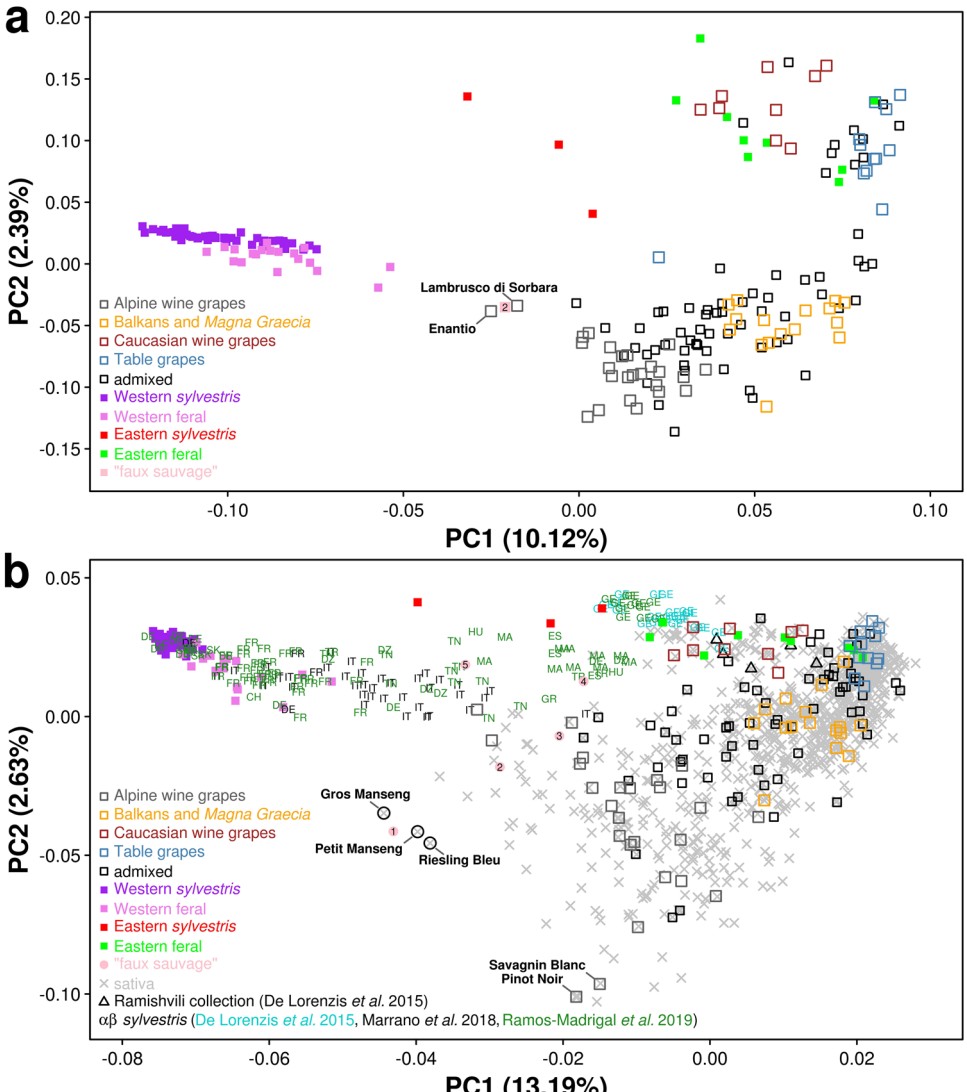

**Fig. 3 Principal component analysis (PCA) in the WGS panel (a) and in the diversity panel (b). a** PCA of 204 *V. vinifera* whole genome resequenced genotypes based on 7.9 M SNPs. **b** PCA of 1445 *V. vinifera* genotypes based on a subset of 6357 pre-ascertained SNPs in the diversity panel and in common with the WGS panel. Sequenced samples are indicated as open (cultivated varieties) and solid (wild accessions) squares. Additional cultivated varieties are indicated as gray crosses in **b**. Samples with uncertain assignment in their literature reports are reported as "faux sauvage": 1, *sylvestris* FR BOOERBY[15]; 2, KE–06[18]; 3, Vigne sauvage faux "Mouchouses 1"; 4, "Tighzirt 1"; 5, "Fethiye 58 64"[15] and collectively indicated as solid circles in **b**. The 2-letter codes (αβ) indicate countries of origin: CH Switzerland, DE Germany, DZ Algeria, ES Spain, FR France, GE Georgia, GR Greece, HU Hungary, IT Italy, MA Morocco, SK Slovakia, TN Tunisia, TR Turkey. Source data are provided as a Source Data file.

propagation and movement, with varieties that have achieved broad or worldwide distribution and others that have largely remained confined in narrow geographic areas. Using a set of 605 cultivated varieties that provided a nearly proportional representation of those in cultivation in each country (Supplementary Table 2), we associated the individual accessions with a precise geographic location represented by either the most ancient known area of cultivation (for widely spread and so-called international varieties) or the most typical or renowned growing region at the present time (for locally grown varieties). Figure 4 shows the geographical distribution of genetic ancestry components for the cultivated compartment (Fig. 4a) and for wild populations (Fig. 4b), respectively. The top two wine-producing countries, France and Italy, exploited most of the diversity of western wine grapes (Fig. 4c, d). The Italian viticulture showed the highest within country variation both in the intensity of the local major ancestry component (Fig. 4c) and in the assortment of

all four ancestry components (Fig. 4d). This was already apparent from the very high proportion of admixed ancestry varieties observed among those originating from the Italian peninsula (Supplementary Fig. 22), which therefore seems to be home not only to varieties that differ in their ancestry but also to crosses that generated highly admixed ancestries. This is likely due both to the historical presence of hubs for maritime and land trade routes between the East and the West and to the ample latitudinal and climatic range of wine growing regions (from 36° to 46°N) that encompass USDA hardiness zones from 7 to 10. Spain and Portugal, the third and fifth wine-producing countries, respectively, instead rely on a national germplasm largely based on high C1 ancestry that is only admixed with C2 ancestry in northern Portugal, Galicia, and southern Pyrenees, presumably as a consequence of massive natural crossing with descendants of Savagnin Blanc (Supplementary Fig. 30). Germany is the fourth wine-producing country in Europe with several wine regions

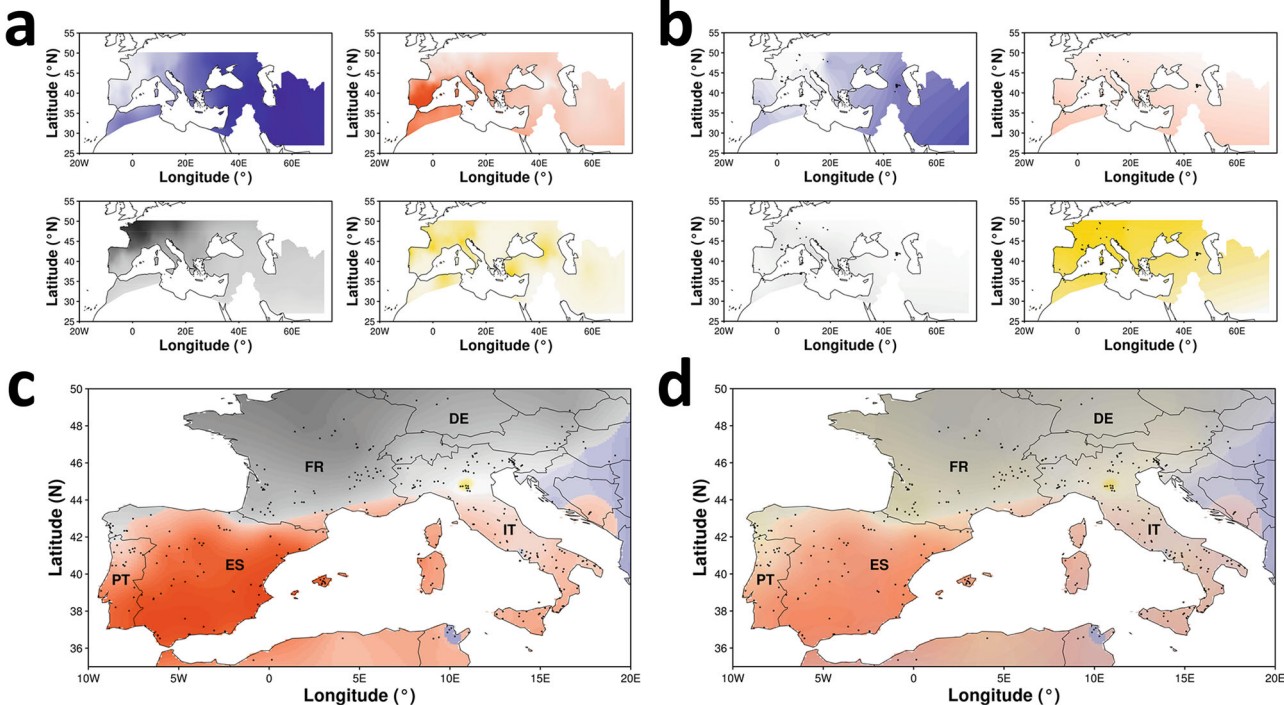

**Fig. 4 Ancestry versus geography.** Continental patterns of ancestry components in cultivated (**a**) and wild (**b**) grapevines and nationwide patterns of wine grape ancestry in the top five wine–producing countries in Europe (**c**, **d**). Colors represent W2 ancestry (blue), C1 ancestry (orange), C2 ancestry (gray), and W1 ancestry (yellow). Each ancestry component is plotted separately (**a**, **b**). Intensity of the main ancestry component is plotted (**c**). Overlay of all ancestry components is plotted (**d**). The collection site of wild accessions is indicated by black dots (**b**). The most representative site of cultivation of each variety is indicate by black dots (**c**, **d**). Abbreviations of top wine-producing countries: Italy IT, France FR, Spain ES, Germany DE, Portugal PT. Source data are provided as a Source Data file.

located at the northern limits of grape cultivation and has, therefore, more limiting growing conditions and a reduced variation in proportional ancestry components across the country. Although there is a clear pattern of ancestry component proportions that is dictated by latitude across the top wine-producing countries, but most notably across Italy (Supplementary Fig. 35), which seems to result from environmental limitations preventing large-scale, within-country geographical displacement, there are notable exceptions of cultivated varieties with typical southern ancestry that are traditionally in use at northern latitudes. For instance, the variety Garganega, once extensively grown in warm climates of Sicily under the synonym of Grecanico Dorato, rose to fame for quality wines only after its long-range movement to Alpine growing regions.

We therefore tested whether local adaptation to climate conditions may have contributed to shaping the geographic distribution of genetic diversity by using a generalized linear model (GLM). For each cultivated variety and the corresponding geographical location, we associated the genetic ancestry coefficients with 29 bioclimatic variables (Supplementary Table 3) of the location using a spatial resolution of $1 km^2$, under the assumption that each variety that has been retained in cultivation in a specific site, where many others may have been discharged, may recapitulate genotypes suitable for the local climate conditions. Seven climatic variables showing <0.70 Spearman correlation with one another explained from 41 to 52% of the variance in the geographic distribution of each ancestry component (Supplementary Table 3). The W2 ancestry showed positive association with annual temperature range and negative association with seasonal precipitation. The C1 and C2 ancestry components showed associations with annual mean temperature in opposite directions (positive and negative, respectively). The

W1 ancestry was most significantly and positively associated with seasonal precipitation. The associations between ancestry components and local climate variables are so tightly related to the geographical location that they rapidly decay under simulations that systematically displaced each genotype outside of the most traditional site of cultivation by 20, 50, and 100 Km in all latitudinal and longitudinal directions (Supplementary Table 4).

**Genomic patterns of non-neutral evolution in cultivated genomes.** Artificial selection for specific desired traits during the domestication process results in selective sweeps that lead to local reductions of genetic variation. Loss of nucleotide and haplotype diversity (Supplementary Fig. 36a) as well as runs-of-homozygosity (Supplementary Fig. 5) were detected in cultivated varieties across three loci on chromosomes 2, 15, and 17 when they were compared to *sylvestris* (Supplementary Fig. 36b, c). These strong signals of selective sweeps presumably originate from strong positive selection of favorable alleles (Supplementary Fig. 36d, f, h) and result in persistent linkage disequilibrium ($r^2$) and extended hitchhiking (Supplementary Fig. 36e).

The reduction of diversity in the lower arm of chromosome 2 is a breeding sweep known to result from positive selection for two nearby loss-of-function mutations causing loss of anthocyanin pigmentation in berry skin[30]. Homozygous recessive genotypes are so-called white varieties, which were the only option for the production of white wines before the advent of modern technologies to limit skin contact of crashed berries with their juice. The quest for this trait brought about a severe loss of diversity at nearby distal loci because, while LD dropped rapidly to background values on the proximal side of the locus, it persisted for 4 Mb on the distal side (Supplementary Fig. 36e).

The reduction of diversity on chromosome 15 resides in a pericentromeric region (Supplementary Fig. 37). Contrary to breeding and domestication sweeps that are characterized by both low haplotype diversity as well as high frequency of homozygous varieties as a result of the positive selection for one favorable mutation, we observed in this case only a marked reduction of haplotype diversity, upstream of the centromere. We also observed an extreme segregation distortion immediately downstream of the centromere in the selfed progeny of Pinot Noir with a complete lack of one class of homozygous seedlings, compatible with a lethal recessive variant that was masked in Pinot Noir by the presence of one copy of the reference haplotype. It is thus possible that favorable and unfavorable variants are in strong linkage and in repulsion across the centromere in this region and are maintained in heterozygous state in the population of cultivated varieties.

The sweep on chromosome 17 has been proposed by Myles and coworkers[1] as a footprint of domestication. Within a 2 Mb valley of haplotype diversity in the cultivated germplasm (Fig. 5a), we identified the nadir of haplotype diversity in a 100 Kb interval carrying a total of 13 predicted genes in the most common haplotype, five of which form a cluster of tandemly arranged isopiperitenol/carveol dehydrogenases (Fig. 5b). In addition to the most common haplotype (H1-A) that corresponds to the PN40024 reference sequence, we identified 18 other haplotypes with minor frequencies in the population (Supplementary Fig. 38 and Fig. 5c). The phenotypic traits that were subject to selection during domestication[31] were presumably related to flower sex determination, with nearby mutations within a sex locus in the upper arm of chromosome 2 involved in the transition from dioecious plants in *sylvestris* to hermaphrodites in *sativa*[32], and to berry morphology, with an increase in berry size and flesh-to-seed ratio going from *sylvestris* to *sativa* that made the grapes more attractive to human consumption and more amenable to wine making, with their genetic determinants still unknown. Quantitative trait loci (QTLs) controlling a series of berry traits in wine as well as in table grapes have been found overlapping with the sweep region on chromosome 17[33,34].

**Characterization of the sweep region on chromosome 17.** Two nearby genes in the sweep region captured our attention because they show the lowest level of diversity (Fig. 5e and Supplementary Fig. 39a) and because they show a hugely increased transcript abundance in the berry in the haplotypes found in the cultivated forms in comparison to those found in the wild ones (Fig. 5f and Supplementary Fig. 39b). These genes (*VIT_217s0000g05570* and *VIT_217s0000g05580*, corresponding to gene numbers 6 and 7 in the diagram of Fig. 5b) are arranged in a head-to-tail orientation with less than 100 bp separating their transcriptional units (Supplementary Fig. 40) and encode a leucine-rich-repeat receptor-like kinase (LRR–RLK) and the first isopiperitenol/carveol dehydrogenase in the tandemly repeated cluster, respectively. We used allele-specific analysis of gene expression to determine the steady-state transcript abundance of the two genes in leaves and berries for a large subset of the 19 haplotypes identified in the region. While no major differences in expression between haplotypes are detected in leaves (Fig. 5f and Supplementary Fig. 39), only one haplogroup, including the most common haplotype and other highly similar haplotypes that are present only in cultivated varieties, seems to produce detectable levels of transcripts in the berries at least for the kinase gene (Fig. 5f). Haplotypes that are found in the wild accessions on the contrary all show transcript levels that are very close to zero. The most frequent (76%) haplotype H1-A is present in 95.8% of cultivated varieties in either homozygous (55.4%) or heterozygous (40.4%) condition, possibly

indicating a dominant or semi-dominant mode of action of the selected allele (Fig. 5d). The haplogroup comprising H1-A is represented in 98.3% of cultivated varieties. The only exceptions among cultivated varieties are represented by Berzamino, an almost abandoned wine grape once grown in Northeastern Italy[35,36], and Gordin Verde, a wine grape from Moldova, unrelated to Berzamino (Supplementary Figs. 24 and 30). Despite both varieties having domesticated traits, Berzamino is homozygous for the H7 haplotype that is predominant in wild accessions and consequently has extremely low transcript levels for both genes in the berry (Fig. 5d–f and Supplementary Fig. 39). Gordin Verde is heterozygous for two haplotypes (H1-F/H2-A) that are normally found in other cultivated varieties in combination with H1-A and provide low transcript levels for the kinase in the berry (Fig. 5d–f). The haplotypes found in all other domesticated varieties that do not have at least one H1-A copy all share a region of sequence identity that comprises the 5′ intergenic region of the kinase gene, the kinase and the dehydrogenase genes, forming the H1-A haplogroup, and have high levels of expression of the kinase gene in the berry (Fig. 5e). The difference in organ-specific and allele-specific expression is even more dramatic for the isopiperitenol/carveol dehydrogenase with extremely high levels of transcript being detected for the cultivated haplotypes in the berry (Supplementary Fig. 39). While there is in general a good correlation between transcript levels of the kinase and the dehydrogenase genes as if there was a common regulatory element capable of affecting the expression of both genes, there are a few haplotypes identified in cultivated varieties (H2-A, H3, and especially H1-F) that show very low levels for the kinase transcript (Fig. 5f) but detectable levels of the dehydrogenase transcript (Supplementary Fig. 39). The expression pattern of the two genes in the berry provided by the selected haplotypes appears to be tightly developmentally regulated (Supplementary Figs. 41–42). Expression is low during the initial phase of berry growth, which occurs mostly by cell division and partly by cell enlargement[37], and sharply increases at berry softening, which marks the inception of ripening about one week before color transition (*véraison*) and resumption of berry growth[38]. This second phase of increase in berry size, unlike the first one, occurs exclusively by cell expansion[39,40]. A genome-wide association study (GWAS, Supplementary Fig. 43) and an association analysis performed with one of the SNPs that recapitulates the expression differences among haplotypes for the kinase (Fig. 6) reveal a significant association between SNPs in the locus and the seed-to-berry ratio at the inception of ripening, with all cultivated varieties showing lower ratios than the wild ones, and Berzamino and Gordin Verde showing high ratios among the cultivated ones. Seed development is the chief factor promoting berry growth[41]. Berry weight, which is commonly measured as a proxy for berry size, is positively correlated with seed content (seed fresh weight, SFW) and QTLs for extreme variation in berry size and seed content colocalize[42] on chromosome 18 with a seed morphogenesis regulator MADS-Box gene[43]. Doligez and coworkers[34] showed, additionally, that the QTL overlapping with the sweep region on chromosome 17 explains the residual variation in berry weight not explained by seed content and it is therefore possible that factors in this region promote pericarp growth at a rate that is more than proportional to the increase in SFW, which is reflected by lower seed-to-berry ratio. The selected haplotypes are associated with a change in berry morphology towards a larger pericarp per unit of SFW. This leads to an increase in size of the fleshy and edible part of the berry, making it more attractive for fresh consumption. It also decreases more than proportionally the seed content released from crushed berries into the must, which greatly improves tannin chemistry and textural sensory attributes in wines. This

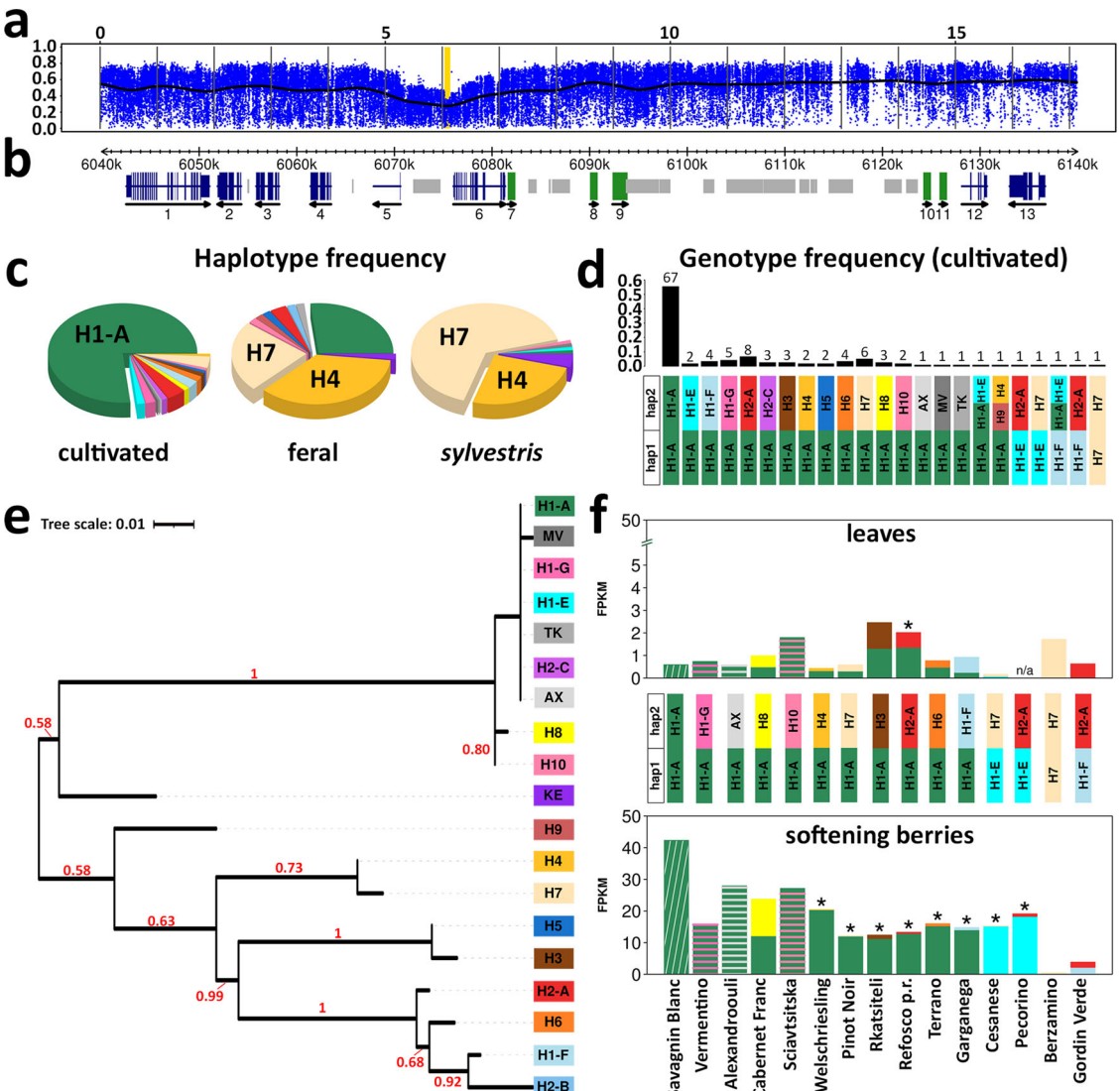

**Fig. 5 Selective sweep on chromosome 17 and allele specific expression (ASE) of the LRR–receptor kinase *VIT_217s0000g05570*. a** Chromosomal plot of haplotype diversity. Haplotype diversity was calculated in blocks of five consecutive variant sites and plotted as the average of 50 consecutive blocks (blue dots) and a cubic smoothing spline (black line). The scale indicates Mb. The yellow background indicate the interval magnified in **b**. **b** V2.1 gene models (exons in blue), manually curated gene predictions (green) in the isopiperitenol/carveol dehydrogenase gene cluster (gene IDs 7 → 11), annotated transposable elements (light gray). **c** Frequency of 19 haplotypes shown in Supplementary Fig. 38 in 196 grapevine accessions. **d** Genotype frequency in 121 cultivated varieties. **e** *VIT_217s0000g05570* (gene 6 in **b**) gene phylogeny. Numbers indicate the proportion of bootstrap trees supporting that clade. **f** ASE of the LRR–receptor kinase *VIT_217s0000g05570* alleles in representative varieties of 15 haplotypic combinations, in softening berries (lower panel) and leaves (upper panel). The asterisks indicate statistically significant ASE levels (*p*-value <0.05) according to a Stouffer's meta-analysis with weight and direction effect using $n = 2$ biologically independent samples. Cumulative expression is reported for each haplotypic combination lacking exonic SNPs in *VIT_217s0000g05570* (H1-A/H1-G, H1-A/H10, H1-A/AX) and for a control variety homozygous for the H1-A haplotype. Gene expression for three haplotype combinations (H1-A/H10, H1-A/H6, H1-A/H4) was quantified in leaves of three different representative varieties (Tschvediansis Tetra, Picolit, Lambrusco Grasparossa) with the same genotype with respect to those used for berry gene expression. Source data of gene expression are provided as a Source Data file.

effect is due to a reduction of the leakage during maceration of astringent and bitter condensed tannins with low degree of polymerization from seeds in favor of the extraction of more palatable condensed tannins with higher degree of polymerization from skins. The kinase gene encodes a LRR–RLK that is ortho-logous to a kinase in Arabidopsis (*At5G62710*) that is expressed in ovaries and in vascular tissues[44] and that shows high homology with the FEI2 kinase. RLKs play a pivotal role in sensing external stimuli, activating downstream signaling pathways and regulating cell behavior involved in response to pathogens, growth, and developmental processes in plants. The FEI2 kinase has been

shown in Arabidopsis[45] to promote anisotropic cell expansion through a modulation of cell wall function, a role that FEI2 fulfils by interacting directly with 1-aminocyclopropane-1-carboxylic acid (ACC) synthase, a key enzyme for ethylene biosynthesis. The grape berry is considered a non-climacteric fruit, lacking a con-comitant increase in respiration rate and ethylene biosynthesis at the onset of ripening, but the rise in endogenous ethylene pro-duction that is consistently observed a few days before the inception of the second phase of berry growth regulates several aspects of ripening[46], including an increase of berry diameter that can be further augmented by the application of exogenous

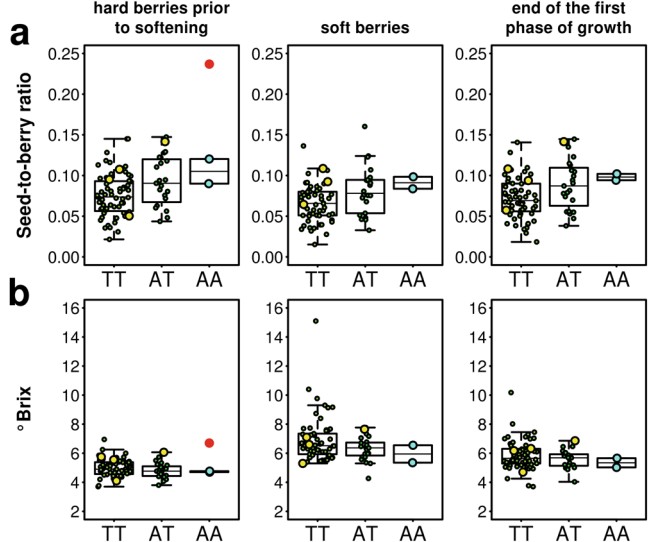

**Fig. 6 Association analysis between SNPs and seed-to-berry ratio at the onset of berry ripening. a** Association between a A → T mutation in the *VIT_217s0000g05570* gene, which recapitulate the increase in berry–specific expression of the kinase, and seed-to-berry ratio in hard berries prior to softening, soft berries collected over the same bunch and their average (as a proxy for the end of the first phase of berry growth). Box-plots show 88 accessions (green dots) sorted by their genotype at the SNP_chr17:6,079,793. Accessions with missing AA, AT, TT genotypes were classified based on their alternate/alternate, alternate/reference and reference/reference genotypes, respectively, at the variant sites chr17:6,080,166; 6,079,793; 6,080,193; 6,080,258; 6,080,447; 6,080,449, which are all in LD with chr17: 6,079,793 in the H1-A haplotype. **b** Variation in soluble solids concentration in the same berries and accessions as in **a**. Red dots indicate values in hard berries of *sylvestris* V395. Yellow dots indicate values in eastern feral grapes. Cyan dots indicate values in Berzamino and Gordin Verde. Boxes indicate the first and third quartiles, the horizontal line within the boxes indicates the median and the whiskers indicate ±1.5 × interquartile range. Source data are provided as a Source Data file.

ethylene at *véraison*[47]. In light of the specific function of the LRR–receptor kinase ortholog in other plants, it is possible that the haplotypes selected during grape domestication may have provided cultivated varieties with new opportunities for ethylene-related cell expansion during berry ripening thanks to the greatly increased expression of the LRR–receptor kinase gene.

## Methods

**Plant material and DNA sequencing**. Details of sequenced accessions are reported in Supplementary Data 1 and Supplementary Note 1. DNA was extracted with a CTAB-based method and fragmented by sonication. DNA libraries were generated according to standard Illumina protocols. Paired-end reads were obtained from Illumina Genome Analyzer II, HiSeq2000, and HiSeq2500 sequencers (Supplementary Method 3).

**Phenotypic data and RNA sequencing**. Two-bud cuttings of 15 cultivated varieties (replicated twice) were forced to rooting in potted soil. A single shoot per cutting was raised until the stage of 10–12 leaves in a common garden experiment. At that stage, the fourth distal (fully expanded) leaf was sampled from each replicate and variety at the same time and frozen immediately for RNA extraction. Berries of an expanded set of 90 accessions were sampled at same developmental stage on different dates according to their curve of berry growth, from two replicated field plots. From each plot two batches of asynchronous berries were collected over the same bunches, one composed of hard berries (target developmental stage: 5.2° Brix), the other composed of soft berries (target developmental stage: 6.4° Brix), both sorted by firmness to the touch. The accuracy of berry sorting was validated by subsampling from each batch random subsets of berries for destructive measurements, e.g., soluble solids concentration (Fig. 6), berry weight, number of seeds per berry, seed fresh weight, and derived parameters (Supplementary Data 3).

The rest of the intact berries were frozen for RNA extraction. RNA was extracted for 15 varieties (replicated twice) using the Spectrum Plant Total RNA Kit (Sigma-Aldrich, Saint Louis, MO). Approximately 500 ng of RNA was used for library construction with the TruSeq Stranded mRNA Kit (Illumina, San Diego, CA) for leaf RNA and with the Universal Plus mRNA-Seq Library Preparation Kit (Tecan Genomics, Redwood City, CA) for berry RNA.

**Variant calling**. Filtered DNA reads (Supplementary Method 4) were aligned to the reference genome using the Burrows–Wheeler Aligner[48]. Raw variants were called using the UnifiedGenotyper tool in GATK[49] version 3.3-0 with 0.01 heterozygosity parameter. Variant sites in each variety were retained if they matched the following parameters: Phred-scaled quality score >50; five reads of minimum coverage; read coverage comprised between 0.5 and 1.5-fold the modal coverage of the variety, maximum coverage relaxed to 2.5 if the SNP passed the filters in other five varieties, minimum coverage relaxed to 0.25 fold the modal coverage in case of homozygous calls; reference/alternate allele coverage ratio between 0.25 and 0.75 for calling heterozygous genotypes, normalized Phred-scaled likelihoods (PL) of the homozygous genotype = 0 and PL value of non-zero PL values > 10 for homozygous assignment; ratio between allelic depth and depth passing quality filters (AD/DP) > 0.05; minimum number of varieties carrying the variant = 1. Variant sites were retained in the dataset if genotype calls passed the filters in >50% of the varieties. Variants sites in repetitive DNA were removed from subsequent analyses. Repetitive DNA included high copy number sequences reconstructed by ReAS[50], transposable elements annotated in Repbase[51], and microsatellites ±10 bp identified using the software Sputnik.

**Gene expression analysis**. Allele-specific gene expression was quantified using duplicated reference assemblies representing the haploid genomes in which, for each cultivated variety, phased SNPs replaced the original variant in the reference genome. RNA reads were aligned using GSNAP[52] version 2019-03-04 and allele-specific RNA reads were counted with ALLIM[53] version 1.1. Allelic imbalance ratio was expressed as the mean of two replicates and the replicates used for assessing the statistical significance using a Stouffer's meta-analysis with weight and direction effect. Allele-specific expression levels were then determined by a STAR[54] version 2.6 alignment of the same RNA reads against the reference genome, which provided the cumulative FPKM estimation for each gene to be proportionally assigned to each allele based on the allelic imbalance ratio.

**GWAS on seed-to-berry ratio**. GWAS was performed using 88 accessions with fully developed seeds (Supplementary Data 3). Seed-to-berry ratio was averaged in each accession between hard and soft stages and between replicates. A total of 6,860,781 variant sites (e.g., polymorphic and informative in >50% of the accessions) were used for GWAS using the -assoc command in PLINK[55] version 1.07. Correction for population structure and cryptic relatedness was obtained applying the genomic control option (–gc). Type-I errors were controlled using a max(T) permutation procedure with 1000 permutations and plotted in Supplementary Fig. 43.

**Reference genome, genome segmentation, gene annotation, and varietal identity assignment**. Nuclear DNA coordinates refer to the *Vitis vinifera* 12Xv0 genome assembly of the strain PN40024 (GCA_000003745.2) with the exception of the upper arm of chromosome 15. We constructed two genetic maps using S1 families derived from Pinot Noir and Schiava Grossa that incorporated 16,358 and 10,179 SNPs, respectively (Supplementary Method 5). We reordered the scaffolds sc_153, sc_107 and sc_45 in GCA_000003745.2 according to evidence of genetic recombination. The order reported in Supplementary Fig. 37 has been used in this paper for drawing the corresponding chromosome diagram.

Gene models refer to the V2.1 gene annotation[56]. Centromeres were identified by blast search of the centromeric repeats[57]. Chromosome-wide analyses refer to a genome segmentation into 2368 non-overlapping windows of variable size containing 100 Kb of non-repetitive DNA. We compared 1445 SNP profiles at 6357 variant sites in common between WGS and SNP chip datasets and generated identity-by-descent estimates using PLINK. Inconsistency in varietal assignment between identical SNP profiles were reconciled by adopting the cultivar name used by Laucou and coworkers[15], corresponding to the accessions held in the germplasm repositories of INRA Domaine de Vassal (France), IMIDRA Finca El Encin, Madrid and ICVV, Logroño (Spain) and JKI Geilweilerhof, Siebeldingen (Germany). The accession TA–6264, previously assigned to *sylvestris* by Liang and coworkers[18], was reclassified as *sativa* Listan Negro. The accession Aramon sequenced by Zhou and coworkers[17] was reclassified as Touriga National. The accessions Turkmenistan2, Azerbajian1, Azerbajian2, Pakistan1 and Pakistan3, previously assigned to *sylvestris*[17], were reclassified as eastern feral based on ancestry component proportions. The accessions Turkmenistan1 and Pakistan2 from the same set[17] were reclassified as interspecific backcrosses and therefore excluded from all analyses. Eastern feral accessions from this study showed parent–offspring (V292 with Korza Erevani; V385 with Ichkimar; V410 with Angur Kalan) or second-degree (V267 with Gyulyabi Dagestanskii and Glera; V294 with Ararati, Kandahari Siah, Tagobi; V389 with Agadai, Ararati, Gyulyabi Dagestanskii, Narma, Tavkveri; V411 with Gyulyabi Dagestanskii) relationships with cultivated

varieties. The pairs K2 and TA–6263, K26 and K27 showed identical SNP profiles. The pairs K22 and TA–6245, KE–23 and TA–6253, KE–23 and TA–6246, Pakistan1 and Pakistan3 showed PO relationships.

**Identification of population structure, signatures of introgression, selective sweeps, haplotype analysis**. Cultivated varieties were assigned to one of the ADMIXTURE K groups based on ≥0.85 genetic ancestry. The other cultivated varieties with proportional membership <0.85 were assigned to admixed populations. Feral or wild western Europe grapes were assigned to western *sylvestris* based on proportional membership ≥0.99 or to western feral otherwise. SNPs in the large sweep region on chromosome 2 were excluded from ADMIXTURE analyses. For TreeMix analysis only, eastern feral grapes were considered as such only if they did not share first-degree or second-degree relationships with cultivated varieties. The ML trees shown in the main text figures were built using blocks of 300 SNPs for the WGS dataset with correction for sample size effects and blocks of ten SNPs for the SNP-chip based dataset. Validation with other combinations of window sizes and number of migration events are shown in Supplementary Fig. 14. Migration weight was expressed as the fraction of alleles donated by the parental population. The variance of relatedness between populations explained by alternative models was calculated using the script treemixVarianceExplained.R (GitHub RADpipe repository, doi: 10.5281/zenodo.17809). ABBA–BABA statistics were calculated in sliding windows of 100 Kb and a step size of 25 Kb, using four groups with the following relationships: outgroup = *Vitis* sp., recipient = Table grapes, donor = western *sylvestris*, generating = Alpine wine grapes. Excess of shared derived alleles between *sylvestris* and Alpine wine grapes was expressed by a modified version of Patterson's D statistics ($f_d$), adjusted to purge signals of introgression from ancestral population structure[25]. Haplotypes in the selective sweep of chr17 were first inferred from heterozygous accessions carrying a reference haplotype. This initial set of haplotypes was used to infer additional haplotypes from other heterozygous accession. Haplotypes could be inferred from 196 out of 204 accession, including 121 out of 123 cultivated varieties. Gene phylogeny reported in Fig. 5 was inferred from maximum likelihood trees based on haplotype Muscle alignments[58] curated by Gblocks. Trees were drawn with TreeDyn. Geographical maps of spatial population genetic structure were drawn using tess3r[59].

**Genealogical relationships**. Pairwise IBD was estimated in 100 Kb windows of non-repetitive DNA (Supplementary Fig. 2), based on thresholds of identity-by-state ratio (Equation 1 in Supplementary Method 6) and genotypic distance (Equation 2 in Supplementary Method 6). Haplotypic distance was calculated based on aggregate lengths of IBD = 0, IBD = 1 and IBD = 2 windows (Supplementary Fig. 33) using the Equation 3 (Supplementary Method 6). The degree of relative consanguinity was assigned based on the aggregate length and distribution of IBD = 0, IBD = 1 and IBD = 2 windows, using the following parameters. PO: IBD = 0 < 15%, IBD = 1 > 50% of the genome length, standard deviation across the genome (st.dev.genome) of IBD = 0 segment length <0.1 Mb. FS: 10% < IBD = 0 > 20%, IBD = 1 < 60% of the genome length, st.dev.genome of IBD = 0 segment length >0.1 and <0.2 Mb, st.dev.genome of IBD = 2 segment length >0.23 Mb, standard deviation of mean segment length per chromosome (st.dev.chr) of IBD = 2 segments >3 Mb. HS: 12% < IBD = 0 > 50 %, IBD = 1 > 60% of the genome length, st.dev.genome of IBD = 2 segment length <0.23 Mb, st.dev.chr of IBD = 2 segments <5 Mb. Clonal variants or duplicated samples: IBD = 0 < 3%, IBD = 1 > 3%, IBD = 2 > 90% of the genome length. Higher than second degree (>2nd degree) under all other conditions.

**Climate data**. Twenty-two monthly, quarterly, seasonal, or annual aggregate values of temperature, precipitation, day length (WorldClim BIO1 to BIO19, min, max, mean day length) were downloaded from WorldClim (www.worldclim.org). Data was sourced from 30 arc-second rasters corresponding to a spatial resolution of 1 km$^2$ at the equator and represent the average of the temporal range 1970–2000. Seasonal potential evapotranspiration (PET) was calculated using the Thornthwaite equation[60] as a sum of the monthly PET from April through September. Seasonal climatic water deficit (cwd) was calculated as the sum of monthly deficit between precipitation and PET form April through September. To avoid the biases due to data multicollinearity among independent variables in the GLM for correlating climate variables with genetic variation, we first identified correlation among climate variables (Supplementary Table 3). Then, seven variables that showed <0.70 Spearman correlation with one another were incorporated in the model where the proportion for one of the four ancestry components was the dependent variable. Linear regression was calculated using unweighted least squares with the lm function in R.

**Reporting summary**. Further information on research design is available in the Nature Research Reporting Summary linked to this article.

## Data availability

The DNA and RNA data generated in this study have been deposited in Sequence Read Archive (SRA) under the BioProject numbers PRJNA373967, PRJNA390884, PRJNA385116, PRJNA321480. The sequences of the reference genome used in this study

are accessible under the BioProject number PRJEA18785. The phenotypic data generated in this study are provided as Supplementary Data 3. Source data are provided with this paper.

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

## Acknowledgements

We thank the grape germplasm repositories of Consiglio per la Ricerca in agricoltura e l'analisi dell'Economia Agraria (CREA), Italy; Institut national de la recherche agronomique (INRA), Unité Expérimentale du Domaine de Vassal, France; Julius Kuhn Institute, Germany; Kmetijsko Gozdarski Zavod Nova Gorica, Slovenia; and Vivai Cooperativi Rauscedo (VCR) Research Center, Italy, Italy for plant material. We also thank Goran Zdunić and Andrey Zvyagin for sharing DNA samples; Paolo Sivilotti for discussions on the variety Berzamino, Andrea Bertoli, Mara Miculan and Nicoletta Felice for DNA and RNA library preparation. This work was supported by FP7–IDEAS–ERC "Novabreed" (grant agreement no. 294780, M.M.), by the Italian Ministry of Agriculture (project Vigneto, M.M.), and by the European Regional Development Fund, Interreg Italy–Slovenia Programme 2007–2013 (grant no 081-3/2011, project VISO, G.D.G.).

## Author contributions

G.M. carried out bioinformatics analyses, I.J. carried out DNA-Seq, A.F. carried out ddRAD sequencing and genetic map construction, E.P. and R.S. carried out RNA-Seq, F.M. carried out statistical analysis, G.D.G. and M.M. conceived the study and wrote the manuscript. All authors have contributed to the revision of the manuscript.

## Competing interests

The authors declare no competing interests.
