## [Peer Review File · Nature Communications]

The genomes of 204 *Vitis vinifera* accessions reveal the origin of European wine grapesReviewers' Comments:

Reviewer #1:

Remarks to the Author:

The authors analyzed the genome of 204 accessions of *Vitis vinifera* collected from Portugal to Tajikistan with the aim to reveal the history and the genetic ancestry of European wine grapes. The topic is interesting and highly debated and the data obtained could be useful to clarify some aspect of domestication. On the other hand, I have observed severe lacks that should be addressed by authors about the evolution of populations (groups), the migrations observed and climatic variations. These problems regard especially the first part of results that include the paragraph named "Population structure and origin of Western European wine grapes". In particular, I believe that the results produced by Treemix are not acceptable in this version. Instead, for the last part of results I have not recommendations, even if I have not personal experience on this issue.

However, I suggest splitting the results and discussion in different paragraphs because I believe that, in this version, the reading and comprehension is complicated. A traditional separation in paragraphs should help also the authors to expose the results clearly and avoid to omit important information. I can understand that the space available in the main text is tiny but, you have a lot of space in supplementary material. Moreover, I suggest reading the wide bibliography about the history and the distribution of grapes and then discuss deeply among the authors before to rewriting the text. The authors will find my comments split in general questions and specific questions below.

General questions:

TREEMIX

Treemix model infers patterns of population splits and mixtures from allele frequency data. It is a simplification of the migration process and, as suggested by authors of software, it may have some limitations if the history of the populations is not largely tree-like. Moreover, flow between populations should be restricted to a relatively short time period. Situations of continuous migration violate this assumption and lead to unclear results. The authors of software underline that in complex structure, there will be many graphs that lead to identical covariance matrices, and thus several different histories will be compatible with the data. Observing your results, I think that your data fall in this situation, but I cannot be sure, because a vast number of information are missing. However, the graph of Figure 1D is particularly creepy. The results in the pie charts, the excessive number of migrations and the short branches indicate that probably the relationships between populations are not tree-like. Moreover, I note also that standard error in figure 1D is longer than several branches. This result could be a problem in the resolution of relationships between populations and the authors should consider this issue. I am not an expert but reading the the manual and the several papers published, confirm more and more my opinions; thus I underline that the authors should not elude these problems. Some specific doubts about the interpretation of migrations are reported in the section Specific questions.

Other important issues that are not explained in your manuscript:

Blocks: The authors of software suggest to account for the fact that nearby SNPs are not independent, group them together in windows of size n SNPs. They recommend using a value of n that far exceeds the known extent of LD in the organism in question. How have you considered this problem?

Trees: How many ML searches were performed? In the text is not clear. You should run a sufficient number of searches and then to compare the results. Have you obtained a single tree or several trees? Have you produced a consensus tree from them? Which tree have you used as starting point to model gene flow?

Migrations: I'm not sure but it seems to me that in Treemix you can explore an infinite number of migrations but if the number of admixed populations is large relative to the number of unadmixed populations, the assumption, that the populations are not largely tree-like, breaks down. It is important that you consider these questions when you chose the number of migrations. For examples, in figure 1 seem that you have found an excessive number of migrations. Have you explored the effect of multiple migrations? How change the variance explained between the ML tree without migrations and ML tree with migrations? I suggest adding migrations sequentially and then to observe the

percentage of variance explained. This observation should help you to choose the number of migrations.

Moreover: 1) InTreemix are implemented three- and four-population tests. These tests should be useful to confirm the migrations but I think that you have not used this option 2) For judging the confidence in a given tree topology, you should generate a bootstrap replicate. You should report the bootstrap values 3) You have populations with different number of samples; Have you used a correction for sample size effects? 4) Have you produced the covariance matrix? Have you produced the standard errors for each entry in the covariance matrix? 5) How have you chosen the position of the root? 6) You should report the weight for each migration. 7) Some of your groups are based on a restricted number of samples. This fact can influence the frequencies observed, and thus the results. How have you considered this issue?

GROUPS

I agree that the method of grouping of the accessions is a fundamental step in your work, but I don't understand, because you have used different grouping methods for different analysis (e.g. Treemix, PCOA, Analysis of climatic variables). This fact makes difficult to understand your manuscript. The authors of treemix explored the relationships between human populations based on the sorting on geographic origins and for the dog on breed type. Moreover, I don't find a link between Introduction and rest of manuscript. I am surprised that in Introduction you propose a classification based on 2 groups for sylvestris accessions and 3 groups for sativa accessions and then you have used different methods of grouping. The grouping that you proposed in the Introduction are widely accepted from several researchers. Why have you added new methods of grouping? Anyway, I expected to find online a list including the origin and geographic coordinates for each accession, and a column with the group of affiliation. Moreover, the origin of some groups is not clear, (e.g. Muscat & Danubian, primitive varieties, Alpine wine grapes). These groups should be excluded because ambiguous. For examples, the classification of Alpine grapes could be viewed with suspicion by both agronomists and botanists.

CLIMATE DATA

The paragraph should be rewritten including the complete list of settings used. I can suppose that the authors have used SDMs, but it is not reported in the text. One of the assumptions of the SDMs is that the species is in balance with the climate, or at least that the climate is the main determinant of distribution. This is often not true for cultivated species. Clearly, a species cannot live outside its niche but for cultivated plants the human activity alters the environmental conditions favoring a species in extreme climatic conditions, where it would not grow because it is less competitive than other species. How have you considered the bias of cultivated plants? Moreover, 4.5 km² seems to me a very large cell. In the plains the climate may not change for tens of km but in the contact area between the mountains and the plains, the climate could change suddenly. You should consider this aspect and to use a different value (for example 1 km²), because high mountains are included in your study.

Specific questions:

Abstract: I avoid commenting on the abstract because, I believe that it should be rewritten on the basis of new results.

44. I suggest to use a correct and appropriate botanical nomenclature.

45-46. This sentence needs a citation. ".....a broad geographic distribution consisting of small isolated populations that are scattered across Europe, northern Africa and Western Asia."

54-58. I don't understand why you report these data here; they appear disconnected from context. I suggest to delete this sentence from Introduction and to use this information to discuss your results.

59-64. I agree that the classification of wild and cultivated grapes is all but simple, however, is important to report where you have found it. In the text, you affirm that sylvestris var. aberrans should be almost extinct. Since on this sentence you base a part of your discussion (lines 148-150), I ask you to explain better this affirmation. You should report where you found this information.

Moreover, in some regions, the two forms cohabit. A detailed description of both forms, indicating the areal of diffusion, should be reported.

65-66. "Additionally, botanists have also been termed as feral". I agree with this sentence, but you should move it from here to below (line 70-71) where you explain the feral grapes. Moreover, a citation is needed.

69-70. I do not understand this new grouping. If you recognize the orientalis and pontica groups (properly), why do you have proposed a new group (pure Eastern ancestry)?

70-71. As you wrote here, it seems that feral grapes are present only in the cradle of domestication. Later you introduce the western feral grapes. You should modify the sentence.

75. Myles et al., (2011) don't discuss about the origin of lambrusques, thus You should delete the part of sentence between brackets (line 75). As it is written seems referred to the hypothesis of Myles (2011).

82. In Arroyo-García et al., (2006) analysis of plastid DNA are applied to resolve the origin of grapevine. In the text is explained clearly the hypothesis that the authors have tested. I suggest to go back to the original reference of the hypothesis.

96. Although to determinate the existence of the bottleneck is a principal goal obtained by authors, I am surprised that results, graphs or values are omitted in the main text. Bottleneck in grapevine was observed in several papers. I look forward to seeing a discussion that also involves previously published articles on this topic.

120. I don't understand why you show the Supplementary Figure 4. This analysis seems so far from the aim of manuscript. Although your sampling could be adequate for a populations study, it seems inadequate to study the divergence times between species. However, too few information about how you have conducted the analysis are reported in the text.

145. Why have you choose $K=4$ respect to $K=2$? In the text is not explained and I am surprised that I have not found the likelihood values in the supplementary material. I believe that this choice is a fundamental step because it can strongly influence the results and consequently the discussion. I attend that you report more details and explain better your choice. Observing the little information reported, I would have chosen $K = 2$. Generally, it is better to avoid adding more groups unless their existence can be justified by a corresponding increase in the likelihood.

156-157. The authors said: "These two components could be attributed to the aberrans forms of sylvestris and deserve special attention to better understand the origin of wine grapes." This sentence is not clear. I don't understand why you say that the two additional ancestral components (orange and gray) can be attributed to the aberrans forms of sylvestris. Can you explain how you infer this fact from the results?

164: In figure 1B I don't find the "modern Western wine grape" group that you describe in the Text. What do you mean? Maybe that you mean a clade. If it is so, can you add this information in figure.

165. You said: "...modern Western wine grapes deriving from introgression of western sylvestris individuals into table grapes", I don't see in figure a migration from western sylvestris to table grapes. What am I missing? I see an intense migration on the tip of Alpine wine grapes and a weak migration on the common ancestor branch of the clade (Alpine wine grapes and Balkans and Magna Graecia).

174. I could agree on the geographical division of the groups for sylvestris but you should insert in excel file the geographical origin of each accession (including geographic coordinate) that you have analyzed.

176. Why have you excluded varieties with highly admixed ancestry? In my opinion, these accessions should be the more interesting in a work that attend to explore ancient migrations between populations. I am surprise of this choice.

178. How you have assigned each group to a geographical area of origin or cultivation is not clear. You should explain better your choice. I underline that, for several varieties could be complex to assign a geographical area of origin but I remark that this choice is essential because forming groups with different criteria will lead to different results. Moreover, if I have understood correctly, so far you have divided several groups by geographical area. How do you justify the origin of other groups? for example, primitive European varieties group?

204-208. I agree only partially with your sentence, (see between brackets the comments). You said: "TreeMix also suggested the occurrence of more ancient events of migration within the continental population of sylvestris (so far, it is clear), likely predating domestication and resulting from the Last Glacial period displacement (this affirmation is vague because you have not inferred divergence times

on nodes), with evidence of southward and eastward migration from Northern Europe into the Mediterranean and Balkan populations (arrows from Northern Europe are multi-directional. you should discuss the single migrations. I suggest to assign a letter to each migration and then discuss each migration) and from Southern Europe into Easternmost populations (migrations from glacial refugia are widely discussed in several papers for animals, plants and also grapes; these last migrations that you describe from Southern Europe into Easternmost populations are a novelty for me. I think that you should be more precise; perhaps you are confusing the effect of glaciation migrations with the spread of varieties through humans).” You should rewrite this sentence.

209-2011. Also, I agree only partially with your sentence (see between brackets the comments): You said: “The scenario depicted by the population structure and relationship analysis presented here is fully consistent with the arrival of table grapes from the East to the Mediterranean Basin (I agree with you because the origin of table grapes is clear in Figure 1) followed by the occurrence of multiple hybridization events with different populations of local wild grapes (This second assertion conflict with you results; In figure 1B, I see only a flow between Table grapes and Eastern feral, but Mediterranean wild grapes are not involved; In figure 1C, I don’t see any migration).” You should rewrite this sentence.

239-241. Can you deduce the origin of western wine grapes using a PCOA? I suggest to test your hypotheses using specific approaches such as Bayesian methods (they are proposed in several papers). Moreover, I have some questions about Figure 3. The interpretation of Figure is complex, and the symbols are too similar between them (including, also, the colors). The names of groups are different from Figure 1, this is an obstacle in the interpretation of grapevine origin. I suggest to use the same groups in both figures. Moreover: Why the accessions analyzed in this paper are only 4 (green square with cross)? What do you mean for admixed group? What do you mean for feral Adriatica and faux savage? These groups should be explained.

244-248. I am surprised that you want to explain the origin of Lambrusco of Sorbara observing a PCOA that explain the 10% of variants. I think that you should use a different approach. Can you propose a test to verify the origin of Lambrusco? Moreover, I think that you should take into account the origin of family. Can you show where fall the other Lambrusco samples in PCOA? Instead, as you wrote, it just sounds like a speculation. However, the topic is interesting, I have found several papers focus that suggest the origins of Lambrusco family, placed between sylvestris and sativa. You should highlight what your results add to the previous studies.

248-249. Hybridization events within the autochthonous sylvestris populations from Ketsch are well known. You should highlight how your results add information to the previous studies.

291-310. In these lines I read a miscellaneous of methods and results about the study of climatic variables. I suggest to split these informations in two distinct paragraphs. However, the results are not used to explain the diffusion of grapevine. Why have you applied these analyses?

I expect that you to discuss your results in a context of origin and diffusion of the grapevine. Other studies on this topic are proposed. I have added the title below; you should consider them to prepare a complete discussion.

513-516. I don’t understand why you have used two different proportional memberships to assign the admixed populations. If I understand correctly, you have used 0.85 as cut-off for cultivated varieties and 0.99 for wild plants. I believe that you should use the same method.

516-518. This choice is not clear: “For TreeMix analysis only, Eastern Feral grapes were considered as such only if they did not share first- or second-degree relationships with cultivated varieties.”

518. Why have you used blocks of 300 SNPs? How have identify the appropriate block size? Have you applied a specific test to make this choice?

520. I have not found the tables that report the ABBA-BABA results.

521. This choice is not clear: “ancestral = Table grapes”. Why have you used Table grapes as an ancestral group? Many of your assertions suggest that table grapes are of recent origin. I report only some your assertions: Line 50: Table grapes were also introduced eastwards into Central Asia along land trade routes. Line 165:introgression of western sylvestris individuals into table grapes. Line 198: ...wine grapes and table grapes have diverged for 2.6 K years. Why do you consider the table grapes as ancient? Wouldn't it be better to use sylvestris?

522. What do you mean for “Alpine wine grapes”. I think that you mean sylvestris.

529. How have you produced the alignment by Muscle (including number of characters) and how have you filtered the alignment by Gblocks? These data are not reported.

FIGURE 1: Number of samples on pie charts should be reported for each group. I don't understand the pie charts in the black box; sativa and sylvestris? Can you explain better?

My considerations are based on personal experience and the follow bibliography.

Geographic distribution of grapevine.

Laocucu et al., (2018) PLoS ONE 13(2): e0192540.

De Lorenzis et al. (2019) BMC Plant Biology, 19:7.

Arroyo-García, R. et al. (2006). Mol. Ecol. 15, 3707–3714.

Climate data

Araújo et al (2019). Science Advances 5: eaat4858.

Zurell et al., (2020). Ecography, 43: 1261-1277.

Garzon et al., (2019) New Phytol, 222: 1757-1765.

Lambrusco and Ketsch populations

Emanuelli, F. et al. (2013). BMC Plant Biol. 13, 39

Schneider et al., (2009) Acta Hort. 827

Grassi et al., (2003) Theoretical and Applied genetics. 107:1315-1320

Schroder et al., (2015) Can. J. Plant Sci. 95:1-8

Treemix

Pickrell, J. K. & Pritchard, J. K. (2012). PLoS Genet. 8, e1002967

Manual for TreeMix v1.1 Joseph K. Pickrell, Jonathan K. Pritchard October 1, 2012

Several papers in which the authors have used Treemix on plants.

Reviewer #2:

Remarks to the Author:

This ms by Magris et al. is a follow-up of the paper by Liang et al. (2019) on genome re-sequencing of 472 grapevine accessions. Liang et al. included most of the *Vitis* species and a number of *Vitis vinifera* accessions in their study and their work provided interesting data on *Vitis* phylogeny, on cultivated and wild *V. vinifera* population structure and demography, on identity-by-descent relationships among cultivars, on genes targeted by artificial selection during domestication. However, they acknowledged that their dataset did not include enough European grapevines (both wild and cultivated) to really explore the history of cultivated grapevines and the introgression with their wild counterparts in Europe. Magris et al.'s study is an attempt to do this. Conceptually, both studies are quite similar, the major difference lies in the dataset. Magris et al. focused on the European accessions and they sequenced 124 new accessions (and analysed a total of 204 ones). They performed population structure and demographic analysis, cultivated grapevine pedigrees, GWAS, selective sweep analysis. These are the analyses that I found particularly interesting:

1- They studied introgressions among cultivated and wild European grapevines, and found that these introgressions have been pervasive, which explains why the European cultivars are different from other cultivars. They concluded that their data do not support a secondary domestication event in Europe (but see my comment below).

2- They analysed the distribution of the wild and cultivated grapevines and found correlations with some bioclimatic variables. A similar analysis was not in Liang et al. and provided some interesting data.

Re-sequencing 124 accessions is less than 472 in Liang et al. but it is still a fairly impressive sequencing effort and large dataset. They provided important information on European grapevine cultivars, complementing Liang et al.'s paper. I think this ms merits a publication in Nature

communications pending some revisions:

1) One key missing information is the geographic distribution of the 124 (and even 204) accessions studied here. There is an excel file in the Supplementary Material showing the list of studied accessions, but this list does not include the country of origin of the accessions. It is currently difficult to have an idea of how well the grapevine diversity (in particular the European one) is represented in the studied sample. This is very important as it can affect the conclusions. For example, the authors have tested the idea of a secondary domestication event in Europe. The secondary event was proposed to be located in the Iberian Peninsula, possibly in Portugal (their ref. 8). Sampling in this area is thus critical. Other analyses presented in the ms may be affected. The information about the country of origin should be added in the excel file. A figure with a map of Europe locating the studied accessions should also be provided. A table showing how much of the grapevine diversity of the main wine-producing European countries was studied here (simply computing the number of studied accession over the total number of accessions in a country) would also be very helpful.

2) I found the first parts of the Results and Discussion very interesting. I was less impressed by the last section (on non-neutral evolution), and found this section was a bit long. I think the ms could be streamlined and written in a more straight-to-the-point way, the letter format may be even more suitable than the article one. In this case, Figures 5 and 6 could go to Supplementary material.

3) P3, lines 107-114: it would be useful to add information on the quality of the data: % of mapped reads, estimated mapping error rates, genome coverage.

4) P3, lines 115-120: same as comment 3 for the SNPs. Please, give details on filtering and comment on the T_i/T_v rates.

5) P4, lines 130-132: "it still clearly shows that the domestication events that led to the creation of the cultivated varieties, unlike in other crops and fruit trees^{18,19}, did not lead to significant genome-wide losses of genetic diversity as a consequence of a major genetic bottleneck." Perhaps the authors could elaborate a little bit more and discuss and cite other papers, for example:

Qi J, Liu X, Shen D, Miao H, Xie B, Li X, Zeng P, Wang S, Shang Y, Gu X, Du Y, Li Y, Lin T, Yuan J, Yang X, Chen J, Chen H, Xiong X, Huang K, Fei Z, Mao L, Tian L, Städler T, Renner SS, Kamoun S, Lucas WJ, Zhang Z, Huang S. A genomic variation map provides insights into the genetic basis of cucumber domestication and diversity. *Nat Genet.* 2013 Dec;45(12):1510-5. doi: 10.1038/ng.2801. Epub 2013 Oct 20. PMID: 24141363.

Cubry P, Tranchant-Dubreuil C, Thuillet AC, Monat C, Ndjiondjop MN, Labadie K, Cruaud C, Engelen S, Scarcelli N, Rhoné B, Burgarella C, Dupuy C, Larmande P, Wincker P, François O, Sabot F, Vigouroux Y. The Rise and Fall of African Rice Cultivation Revealed by Analysis of 246 New Genomes. *Curr Biol.* 2018 Jul 23;28(14):2274-2282.e6. doi: 10.1016/j.cub.2018.05.066. Epub 2018 Jul 5. PMID: 29983312.

6) In different analyses, the generation time (GT) was set to 3 years. This seems a very small GT for grapevines. Wild grapevines are usually sexually mature after 5 years, and reach optimal reproduction after 7-10 years, some individuals can be >100 years old (see their ref. 10). The impact of GT's value on their conclusions needs to be discussed.

7) I found Supplementary figures 9 and 10 a bit hard to understand. I know that this kind of analysis can return complex networks. But here, most of the connections between the accessions are simply intractable. Simplifying the figure would help. Naming clusters and using different colours for different clusters would also help.

Reviewer #3:

Remarks to the Author:

This submission by Magris and colleagues describes their impressive efforts to generate a massive amount of sequencing data on wild and cultivated *Vitis vinifera*. The data in the study will undoubtedly be used for by grapevine researchers for years, and it has relevance for evolutionary questions as well as for germplasm management. The authors' interpretations on grapevine domestication are also useful, although some of the findings have been made before, albeit on more limited datasets (e.g., SNP chips or whole genome sequencing of few individuals). Still, it is important that genome-wide

data confirm previous understandings of a single domestication origin for cultivated grapevine and a limited genetic bottleneck.

I found it helpful to see this study as a counterpart to the Liang et al. (2019) paper, suggesting *Nature Communications* is a fitting journal for publication. The Liang et al. paper has more grape accessions, but their interest on different *Vitis* species means this paper has a different focus. Overall, I find Magris et al.'s paper to be of a high standard, but there are a few items that could be developed to improve the study.

The most unexpected element of the study is the controlled garden experiments and related transcriptomics. From the abstract, it was unclear this is even a part of the project—it seemed it was only a genomic dataset. It would help to clarify this from the start, as it does give this project a different angle than that of Liang et al., Myles et al., Zhou et al., etc. In the main text, these experiments and the RNA data are mentioned in the very long section “Genomic patterns of non-neutral evolution in cultivated genomes”. This section needs some further clarity and organization there is a single paragraph from lines 339-430. To heighten the reader's comprehension, I suggest creating a new section, but at minimum implement some paragraphs and consolidate some of the discussion on genes and berry size. The alternative option would be to remove the RNA component and expand genomic analyses on some of the other areas of the genome with high levels of homozygosity. The section also ends on with the authors being “tempted to speculate”, which is an unusual tone for a scientific work. I recommend rephrasing into more a more empirical statement, or potentially writing as an open hypothesis to guide future research.

One shortcoming of the study is the emphasis placed into the four ancestral groups. While there clearly is geographic structure in *V. vinifera*, the four groups are often treated like they are biological certainties. Indeed, the cross validation error plot shows a number of very similar values, and $K=3$ looks potentially better than $K=4$. Other researchers have used other values of K , like Bacilieri et al. (2013 *BMC Plant Biology*) using 3 main groups with an additional analysis of $K=5$ to evaluate structure within groups. At the least, the Admixture plots should be presented with several values of K for transparency sake (can be shown in supplemental document if needed). That is a common approach in the field, as exemplified by the Liang et al. study.

Figure 1 needs to be improved in a number of ways. First, panels A and C should have more labels to allow readers to interpret samples/regions. Only Western *sylvestris* and western feral are labeled, and the key in panel C is very difficult to see. I do not understand why panel A has two red areas and two black areas, apparently for wild and cultivated, respectively. These figures may need to be included in a larger format as supplemental figures will proper labels. In panel D, the TreeMix tree has some very strange features, like the Caucasian wine grapes being ancestral to the wild Caucasian grapes. This suggests something is wrong in the model, perhaps overfitting on a small number of datapoints or allowing too many migration events. At the least there should be more discussion of why this may not meet readers' expectations.

One of the goals of the project is stated to be associating genetic variation with climate. As far as I can tell, this was primarily done through by comparing environmental variables and latitude vs. the ancestry of four assumed ancestral groups. The problem is that this greatly depends on the ancestral groups being biological realities rather than the clusters of similar samples found in Admixture software. Presumably individuals within these clusters have different responses to some environmental conditions, and it would be quite intriguing to look at some key variables with a GWAS approach, as you did for berry ripening.

Other points:

In the introduction you write “driven by human migration and maritime trades, paralleled by a

transition in use from fresh consumption (table grapes) to winemaking (wine grapes)". There is no citation for this, and some of the early archaeological evidence from Georgia indicates a very early use of grapes in winemaking (even though one may assume humans have been eating wild grape berries for millennia). Please update the question of table vs. winemaking based on the literature.

For Figure 3, the color pallet makes it difficult to differentiate many of the points. It would help to also vary the shape of the points in addition to the color.

Was any attempt made to look at the indels or transposable elements?

Did you correct for LD? This seems to be particularly important when cultivars have had limited sexual reproduction in the past centuries.

In line 139, you don't specifically mention the analysis, just the authors. To clarify, suggest naming "TreeMix" here so that it is clear when the software is listed on line 163.

Have any of the kinships identified in this study been at odds at previous genomic analyses? That could be an important insight in the importance of WGS vs. the efficiency of SNP chips.

Given that three areas of the genome have been identified as being under selection, this raises the question of how they feature in other analyses. Were the three loci on chromosomes 2, 15, and 17 included in the Admixture and TreeMix analyses? Their inclusion could cause cultivated accessions to appear more closely related than the rest of the genome indicates. It would be interesting to evaluate with both approaches.

I see the data have not yet been released for the main NCBI BioProject. Please ensure they are available at the time of publication.

For Supplementary Figure 3, the three colors are not defined.

For Supplementary Figure 8, you should clarify N_e is the red line. Also, is the generation time of 3 years a realistic value? It surely works before vegetive propagation was commonplace, but I have questions on the reliability of this analysis for the most recent time when some varieties are maintained for centuries.

For Supplementary Table 1, you have used commas for the decimal point, unlike the rest of the article.

Reviewer #1 (Remarks to the Author):

The authors analyzed the genome of 204 accessions of *Vitis vinifera* collected from Portugal to Tajikistan with the aim to reveal the history and the genetic ancestry of European wine grapes. The topic is interesting and highly debated and the data obtained could be useful to clarify some aspect of domestication. On the other hand, I have observed severe lacks that should be addressed by authors about the evolution of populations (groups), the migrations observed and climatic variations. These problems regard especially the first part of results that include the paragraph named "Population structure and origin of Western European wine grapes". In particular, I believe that the results produced by Treemix are not acceptable in this version. Instead, for the last part of results I have not recommendations, even if I have not personal experience on this issue.

However, I suggest splitting the results and discussion in different paragraphs because I believe that, in this version, the reading and comprehension is complicated. A traditional separation in paragraphs should help also the authors to expose the results clearly and avoid to omit important information. I can understand that the space available in the main text is tiny but, you have a lot of space in supplementary material. Moreover, I suggest reading the wide bibliography about the history and the distribution of grapes and then discuss deeply among the authors before to rewriting the text. The authors will find my comments split in general questions and specific questions below.

General questions:

TREEMIX

Treemix model infers patterns of population splits and mixtures from allele frequency data. It is a simplification of the migration process and, as suggested by authors of software, it may have some limitations if the history of the populations is not largely tree-like. Moreover, flow between populations should be restricted to a relatively short time period. Situations of continuous migration violate this assumption and lead to unclear results.

None of the crop species is an ideal population. Nevertheless, the model developed by Pickrell and Pritchard has been used in a large number of diverse applications in plant population genetics. We were aware of the strengths and weaknesses of the model and we are confident to have adopted all measures to minimize the limitations of the predictive model. Support for this claim is provided below and in reply to specific comments.

At the continental and historical scale, we exclude that the condition in the whole grapevine population is a situation of continuous migration (population genetics *sensu stricto*). The condition is not comparable with the crops that are sympatric with endemic populations of their wild relatives and undergo sexual reproduction annually (and by the way, even in those situations introgressions from wild relatives to crops appear to have been an exception rather than a rule), which may support a model of continuous migration over time. One aspect is the extent of migration. A different aspect is the genomic footprint left by these events. We have shown that wild-to-crop migration in Europe occurred more than once, but all evidence points to a limited series of occasional events, which is also compatible with the documented occurrence of only small, isolated and endangered populations of the wild relative in Europe. The results from the vast majority of genetic population papers are also in favor of a substantial separation between grape wild accessions and cultivated varieties, based on population genetics metrics, with very rare exceptions of intermediate forms as well as genetic

footprints of past events of admixture. Regarding the migration events within the cultivated compartment, leading to admixture between different ancestral groups, they were historically due to the occasional introduction of accessions by trading, which is a different condition from continuous migration between sympatric populations as it occurs in annuals.

In the revised version, we provide additional support for the results produced by TreeMix.

The authors of software underline that in complex structure, there will be many graphs that lead to identical covariance matrices, and thus several different histories will be compatible with the data. Observing your results, I think that your data fall in this situation, but I cannot be sure, because a vast number of information are missing. However, the graph of Figure 1D is particularly creepy. The results in the pie charts, the excessive number of migrations and the short branches indicate that probably the relationships between populations are not tree-like.

We provide in the revised version of the main text and in Supplementary Information, the analysis of residuals between different model predictions and actual data, the fraction of the explained variance by different models, and the entire series of tree options that were generated for an increasing number of migration events. We show with this extended data that the relationships between groups and the predicted migration events are consistent across different parameter sets used for data analysis (Supplementary Figures 13–20).

Moreover, I note also that standard error in figure 1D is longer than several branches. This result could be a problem in the resolution of relationships between populations and the authors should consider this issue. I am not an expert but reading the the manual and the several papers published, confirm more and more my opinions; thus I underline that the authors should not elude these problems.

The standard error is much shorter than all branches (Figure 1a and Supplementary Figures 13–20) when TreeMix analysis is performed with the amount of variation captured by WGS data and with the group composition that resulted from ADMIXTURE analysis.

We are aware that lower strength and higher standard errors need to be accepted in exchange for generating a similar output from geographical based clustering (as the one we provided in Figure 1b) which is aimed at addressing questions usually raised by the grape community. The inherent limitations with this TreeMix output (Figure 1b) is due to the lower depth of variation captured by SNP chips and to the use of groups that are defined by geographical distribution of the individuals. However, we are confident with presenting and discussing the results of Figure 1B, because:

- 1) The tree in Figure 1b has maintained the same broad structure of the more robust tree in Figure 1a**
- 2) The introgression events suggested by the models in Figure 1a and 1b are also supported by independent analyses (i.e three–population test and ADMIXTURE ancestry components; Supplementary Figures 7 and 10, Supplementary Tables S1-S2)**

Some specific doubts about the interpretation of migrations are reported in the section Specific questions.

Other important issues that are not explained in your manuscript:

Blocks: The authors of software suggest to account for the fact that nearby SNPs are not independent,

group them together in windows of size n SNPs. They recommend using a value of n that far exceeds the known extent of LD in the organism in question. How have you considered this problem?

This issue was addressed by two approaches.

In the first approach using the complete SNP dataset, we tested all combinations of different block sizes (as suggested by the software developers, in our case from 100 to 10,000 SNPs window sizes) \times the number of migration events (from 1 to 5).

The outputs of all different approaches are included for the readers' convenience in the revised version of Supplementary Information, which show consistency with the representative case shown in the main text.

In the second approach, we considered that any fixed window size (as expressed in terms of number of SNPs included) might be affected by intrachromosomal variation in SNP density and LD levels, which is particularly large between pericentromeric and subtelomeric regions. We therefore pruned the complete SNP dataset for linkage disequilibrium prior to TreeMix analysis and then fed TreeMix with the pruned dataset. Pruning was performed in sliding windows of 50 SNPs with a 10-SNP overlap and removing SNPs with $r^2 > 0.2$. The LD-pruned dataset consisted of 1,548,295 SNPs.

The use of the pruned dataset explained a lower fraction (92.5 %) of the variance of relatedness among populations with the bifurcating tree. Using this approach, we also obtained lower values of explained variance with modelling one or more migration events, compared to fixed windows of any size.

Considering that the full dataset was, anyway, filtered for variant sites in regions of repetitive DNA prior to ADMIXTURE analysis and considering the rapidity of the LD decay in grapevine, it appears from all these tests that, under the case of this crop species and using this procedure, the predictive models generated by TreeMix are robust and resilient to different data treatments.

Trees: How many ML searches were performed? In the text is not clear. You should run a sufficient number of searches and then to compare the results. Have you obtained a single tree or several trees? Have you produced a consensus tree from them? Which tree have you used as starting point to model gene flow? Migrations: I'm not sure but it seems to me that in Treemix you can explore an infinite number of migrations but if the number of admixed populations is large relative to the number of unadmixed populations, the assumption, that the populations are not largely tree-like, breaks down. It is important that you consider these questions when you chose the number of migrations. For examples, in figure 1 seem that you have found an excessive number of migrations. Have you explored the effect of multiple migrations? How change the variance explained between the ML tree without migrations and ML tree with migrations? I suggest adding migrations sequentially and then to observe the percentage of variance explained. This observation should help you to choose the number of migrations.

100 iterations were performed before adding migrations. The support of each branch before modelling admixture events is now indicated in Figure 1 by bootstrap values. Then, migrations were added sequentially and their support was evaluated by the curve of explained variance of the relatedness among populations. We provide now in the revised version of Supplementary Information the entire series of tree options that were generated for an increasing number of migrations.

Moreover:

1) InTreeMix are implemented three- and four-population tests. These tests should be useful to confirm the migrations but I think that you have not used this option

We have performed the three–population test and included it into the results in the revised version of Supplementary Information, where it confirms previous results.

2) For judging the confidence in a given tree topology, you should generate a bootstrap replicate. You should report the bootstrap values

Bootstrap values were added to tree branches.

3) You have populations with different number of samples; Have you used a correction for sample size effects?

Yes, we have.

4) Have you produced the covariance matrix? Have you produced the standard errors for each entry in the covariance matrix?

Yes, we have.

5) How have you chosen the position of the root?

The analysis was run with no *a priori* assumptions. Trees are unrooted.

To doublecheck whether rooting the tree before adding migrations would have had an impact on the prediction of introgression events, we rooted it either using eastern *sylvestris* or using western *sylvestris*. The predicted migrations were totally unaffected by non–rooting or rooting either way.

6) You should report the weight for each migration.

In both the original and the revised version, the weight for each migration is indicated in Figure 1 by the color of the arrows. The heat map scale for the migration weight was included and explained in the Figure Legend.

7) Some your groups are based on a restricted number of samples. This fact can influence the frequencies observed, and thus the results. How have you considered this issue?

We have used the algorithm option in TreeMix that corrects for the effect of sample size.

GROUPS

I agree that the method of grouping of the accessions is a fundamental step in your work, but I don't understand, because you have used different grouping methods for different analysis (e.g. TreeMix, PCOA, Analysis of climatic variables). This fact makes difficult to understand your manuscript. The authors of treeMix explored the relationships between human populations based the sorting on geographic origins and for the dog on breed type.

Regarding the criteria of grouping for TreeMix analysis:

unlike humans and dogs in which there is a strong relationship between geographic origin, breed type and population structure, in grapevine

- 1) the geographic origin (intended either as the most renowned current site of cultivation or the most ancient area of cultivation based on historical records) for each variety is anecdotal
- 2) breed type may include crosses between two parents belonging to highly differentiated genetic pools resulting into highly admixed hybrids.

Under this scenario, we think that grouping solely based on the information of DNA variation data, with an agnostic approach with regard to other assumptions, is necessary for an unbiased approach to population genetic analyses. An exception to this rule was only done for generating Figure 1b, due to the reasons explained above.

Regarding the reviewer's remark that the sample set is different among (1) PCA and ADMIXTURE (all sample included), (2) TreeMix (admixed individuals removed), and (3) correlation between ancestry components and climate variables (a reduced set of samples), this is due to:

- PCA and ADMIXTURE do not require *a priori* grouping and can handle the entire dataset
- TreeMix requires *a priori* clustering of genetic variation into discrete groups of well differentiated individuals. Admixed individuals were excluded from this analysis
- climate variable analysis could be conducted only with a subset of varieties, yet still large ($n=605$), for which peculiar sites of cultivation could be assigned upon a survey of viticulture literature.

Moreover, I don't find a link between Introduction and rest of manuscript. I am surprise that in Introduction you propose a classification based on 2 groups for *sylvestris* accessions and 3 groups for *sativa* accessions and then you have used different methods of grouping. The grouping that you proposed in the Introduction are widely accepted from several researchers. Why have you added new methods of grouping?

As for *sylvestris*, we did not go beyond the divide between Eastern and Western.

In order to obtain *bona fide sylvestris* with respect to the domesticated forms in the East and in the West, there was the need to feed models with a subset of genuinely wild accessions that were more likely to be free from any gene flow from domesticated forms. This has led us to place the leftovers into two groups of Western feral (defined by the metrics provided by populations genetics analysis) and Eastern feral grapes (defined primarily on the fact that those accessions were spontaneous stocks and secondarily supported by populations genetics metrics). For the sake of caution, both groups were treated separately from *bona fide sylvestris*. TreeMix analysis supported the rationale that accessions in these feral groups may have resulted from cultivated-to-wild gene flow, which was also supported, in the case of Eastern feral forms, by the evidence of first-degree relationships found with cultivated varieties (Supplementary Figure 30).

As for *sativa*, the taxonomy treatment we mentioned in the introduction section is based on an eco-geographical classification done by the botanist Negrul, before the advent of molecular analyses. It may or may not reflect population structure at the genetic level.

Negrul has also introduced a number of *taxa* that may be hierarchical with respect to his three chief *taxa* or part of the main divide. The subdivide within one (*pontica*) of the main three Negrul's eco-geographical groups in *pontica georgica* and *pontica balcanica* that he considered as lower hierarchical *taxa* and which roughly correspond, at the population genetic level, to Caucasian wine grapes (*pontica georgica*) and to Mediterranean wine grapes (*pontica balcanica*) seems not to be actually a lower hierarchical split. We provided independent lines of evidence (i.e. admixture and Treemix analysis, pedigree networks, PCA) that Caucasian wine grapes are isolated from the rest of the wine grape germplasm and Mediterranean wine grapes are genetically more similar to table grapes (Negrul's *orientalis*) than to Caucasian wine grapes (Negrul's *pontica georgica*).

The partial inconsistency with Negrul's taxonomic treatment mentioned above and the reviewers's concerns about the number of reliable K groups are based on whether:

- 1) *balcanica* (as defined by ADMIXTURE) should be considered a higher hierarchical group (by adopting a $K \geq 3$) or a lower hierarchical entity (according to Negrul and with $K=2$ that separates only Eastern and Western ancestry)
- 2) *pontica georgica* and *orientalis* (as defined by ADMIXTURE) should be considered as one ancestral population (according to $K \leq 3$ scenarios) or two ancestral groups (according to Negrul and according to the $K=4$ scenario)

There were no substantial differences in varietal composition for the *occidentalis* ADMIXTURE group under $K=2$, $K=3$ and $K=4$ scenarios and for the *balcanica* ADMIXTURE group under $K=3$ and $K=4$ scenarios (Supplementary Figure 10).

Our choice to present in the main text the result obtained with $K=4$ is supported by the fraction of variance explained for the relatedness between populations and by independent analyses of the distribution of genetic variation for the major components (i.e. PCA) and the relationship networks (Supplementary Figure 24 and 30). Although these analyses capture other aspects of genetic variation, they all clearly show an isolation of *pontica georgica* from the rest of the germplasm and an intermediate position of Mediterranean wine grapes (*balcanica*) between *occidentalis*, on the one side, and table grapes, on the other side, which are both explained only with $K=4$.

$K=2$ and $K=3$ also have another major pitfall with respect to $K=4$. They fail to discriminate between the different contribution of Western *sylvestris* into Western cultivated varieties and of Eastern *sylvestris* into Eastern cultivated varieties. In the Source Data file, the spreadsheet "Ancestry V. vinifera" clearly shows that with $K=2$ and $K=3$ Western ancestry would be mistaken for Eastern *sylvestris* ancestry in Caucasian primitive wine grapes (i.e. Adjaluri Tetri, Ojaleshi, Mgaloblishvili).

In order to exclude that the $K=4$ grouping has had a significant impact of the major conclusions of this study, we tested TreeMix for western *sylvestris* introgression under four alternative scenarios, which are reported now for the readers' convenience in Supplementary Information:

- 1) simulation of a three-population scenario with *pontica georgica*, *orientalis*, *occidentalis* ancestral populations and *pontica balcanica* admixed, according to Negrul's taxonomy treatment

- 2) simulation of a three–population scenario with *pontica georgica*, *orientalis*, *occidentalis* ancestral populations, with an extended *occidentalis* group encompassing most of the Mediterranean diversity (i.e. including *pontica balcanica*)
- 3) considering only one Eastern ancestral population (*pontica georgica+orientalis*) and simulating a three–population scenario with Eastern, *balcanica* and *occidentalis* groups, according to the K = 3 ADMIXTURE grouping
- 4) considering only one Eastern ancestral population (*pontica georgica+orientalis*) and simulating a scenario with Eastern and *occidentalis* ancestral populations, according to the K = 2 ADMIXTURE grouping

The conclusions on (1) a single domestication in the East, (2) the post-domestication introgression in the West and (3) the relationships between groups of cultivated varieties were confirmed regardless of the number of K ancestry components from 2 to 4.

We think that all the evidence that is now included in the Supplementary Information provides strong support for the claims that K=4:

- 1) provides more resolution without compromising the robustness of the conclusions
- 2) generates results that are more adherent with the taxonomic treatment of the cultivated compartment
- 3) is consistent with reliable K number identified by Liang et al 2019

Anyway, I expected to find online a list including the origin and geographic coordinates for each accession, and a column with the group of affiliation.

This information has been added to the Source Data file.

Moreover, the origin of some groups is not clear, (e.g Muscat & Danubian, primitive varieties, Alpine wine grapes). These groups should be excluded because ambiguous. For examples, the classification of Alpine grapes could be viewed with suspicion by both agronomy and botanists.

They are defined without ambiguity in Source Data using the criteria of within–group homogeneous ancestry composition and assignment to broad geographic areas of origin or cultivation.

The names of the groups reflect either the grape type or the geographic area that are peculiar to the varieties that happened to cluster together based on the criteria above.

Muscat & Danubian has been changed into Balkans & *Magna Graecia* for consistency with other figures and for adherence to the criteria above.

Alpine referred to countries, that is the territory of eight countries associated with the Alpine region, as defined by international territorial treaties. For the sake of conciseness, “wine grapes from Alpine countries” has been shortened to “Alpine grapes” after first mentioning. If this phrasing sowed confusion, we would maintain the terms “wine grapes from Alpine countries” everywhere in the text.

The group identified as “Primitive varieties” was defined in lines 310-312 along with illustrative examples of individual accessions and its complete composition is reported in Source Data to avoid ambiguity.

CLIMATE DATA

The paragraph should be rewritten including the complete list of settings used. I can suppose that the authors have used SDMs, but it is not reported in the text. One of the assumptions of the SDMs is that the species is in balance with the climate, or at least that the climate is the main determinant of distribution. This is often not true for cultivated species. Clearly, a species cannot live outside its niche but for cultivated plants the human activity alters the environmental conditions favoring a species in extreme climatic conditions, where it would not grow because it is less competitive than other species. How have you considered the bias of cultivated plants?

We did not use species distribution models because they are designed for other purposes, while we sought to explore relationships between genetic variation and climate variables.

We reasoned that the current most representative site of cultivation for each variety is the result of many factors, including historical events and chance. As an empirical result of a partially adaptive and partially stochastic process of geographic dispersion, each variety that has been retained in cultivation for decades or centuries in a peculiar viticulture site, while many others were discharged, should possess a genetic make-up that makes it broadly suitable for the climatic conditions associated with the location.

We therefore used a general linear model, with no assumptions on predictor and response variables to associate climate parameters with ancestry components found at any location.

As for the settings, linear regression was calculated using unweighted least squares with the *lm* function in R. This information has been added to Mat&Met section in the revised version.

As to the bias the reviewer is referring to for cultivated plants, it may be true that in many conditions crops may perform less competitively in comparison to wild species (but usually for reasons other than adaptation and more related to their crop nature) but we think it is equally true that individual varieties within each crop are usually selected so that they perform best in their specific cultivation environment.

Moreover, 4.5 km² seems to me a very large cell. In the plains the climate may not change for tens of km but in the contact area between the mountains and the plains, the climate could change suddenly. You should consider this aspect and to use a different value (for example 1 km²), because high mountains are included in your study.

All data in this paragraph are updated in the revised version by increasing the spatial resolution from 4.5 squared kilometers to 1 squared kilometer, as requested by the reviewer.

We also provided a simulation showing that systematic shifts in the geographic coordinates that displace the assigned peculiar location outside of the characteristic area of cultivation of each variety increasingly disrupt the observed significant associations between ancestry components and climatic variables.

Specific questions:

Abstract: I avoid commenting on the abstract because, I believe that it should be rewritten on the bases of new results.

The abstract has been rewritten to put more emphasis and clarity on the results.

44. I suggest to use a correct and appropriate botanical nomenclature.

We are aware that there is inconsistency in literature reports both on the use of Open Nomenclature qualifiers and on the naming of one of the two taxa within the species *Vitis vinifera*.

We considered the two taxa at the taxonomic rank of subspecies and used therefore the abbreviation of subsp. that is considered interchangeable with the abbreviation ssp.

There are examples of the use of other Open Nomenclature qualifiers for these taxa, such as the use of species in the plural form (abbreviation: spp.), among the references we cited. The spp. qualifier describes the presence of more than one species of the same genus, whose identification has not been accomplished yet. We therefore excluded using this qualifier.

Regarding the naming of the two taxa, while there is consistency in one subspecies (*sylvestris*), there is variation in the naming of the domesticated subspecies. Some authors use *Vitis vinifera* ssp. *sativa*, other authors (i.e. Myles and coworkers among those cited in this paper) use *Vitis vinifera* ssp. *vinifera*. We decided to go with the first option because it is more commonly used in grapevine reports. However, in order to exclude ambiguity,

- 1) we pointed out in the text that the nomenclature used here (ssp. *sativa*) corresponds to the ssp. *vinifera* naming that the readers may find in other papers**
- 2) we also added the original authority, appearing in parentheses, credited with the first formal use of the name and the name of the botanist who later changed it into the current version**

45-46. This sentence needs a citation. ".....a broad geographic distribution consisting of small isolated populations that are scattered across Europe, northern Africa and Western Asia."

Added.

54-58. I don't understand why you report these data here; they appear disconnected from context. I suggest to delete this sentence from Introduction and to use this information to discuss your results.

We still believe that Lines 54-58 were a logical extension of the statements in Lines 50-54 in the original version and they set the necessary background in the Introduction.

59-64. I agree that the classification of wild and cultivated grapes is all but simple, however, is important to report where you have found it. In the text, you affirm that *sylvestris* var. *aberrans* should be almost extinct. Since on this sentence you base a part of your discussion (lines 148-150), I ask you to explain better this affirmation. You should report where you found this information. Moreover, in some regions, the two forms cohabit. A detail description of both forms, indicating the areal of diffusion, should be reported.

The distinguishing morphological traits between *typical* and *aberrans* as well as their geographical distribution have been described and a citation has been added.

65-66. "Additionally, botanists have also been termed as feral". I agree with this sentence, but you should move it from here to below (line 70-71) where you explain the feral grapes. Moreover, a citation is needed.

Yes, Eastern and Western feral grapes are now mentioned sequentially in the revised text. Citations have now been added in order to support the botanical definition of this group and the use of the term "feral".

69-70. I do not understand this new grouping. If you recognize the *orientalis* and *pontica* groups (properly), why do you have proposed a new group (pure Eastern ancestry)?

While *orientalis* and *pontica* groups are recognized by Negrul as separated taxa based on ecogeographical considerations, we want to point out in the Introduction section that Negrul also considered both of them as the groups with pure Eastern origin. We think that this specification is necessary for the readers to interpret correctly the meaning of the K=2 to K=4 divides that emerged in the Result section from ADMIXTURE analysis.

70-71. As you wrote here, it seems that feral grapes are present only in the cradle of domestication. Later you introduce the western feral grapes. You should modify the sentence.

Yes, Eastern and Western feral grapes are now mentioned sequentially in the revised text.

75. Myles et al., (2011) don't discuss about the origin of lambrusques, thus You should delete the part of sentence between brackets (line 75). As it is written seems referred to the hypothesis of Myles (2011).

Yes, Line 100 and the following ones are now rephrased to make more clear that the *lambrusques* nomenclature for that sort of plant material is after Levadoux.

82. In Arroyo-García et al., (2006) analysis of plastid DNA are applied to resolve the origin of grapevine. In the text is explained clearly the hypothesis that the authors have tested. I suggest to go back to the original reference of the hypothesis.

This sentence has been rephrased in the R1 version, paying extreme attention to the semantics: "According to another hypothesis, cultivated grapes have at least two origins, one in the East and the other in the West, and western wine grapes have had their origin in an independent event of domestication from western *sylvestris*."

This statement has been elaborated following a careful revision of the results in the cited article and, more specifically, of the following sentences extracted from the mentioned article:

"The results suggest the existence of at least two important origins for the cultivated germplasm, one in the Near East and another in the western Mediterranean region, the latter of which gave rise to many of the current Western European cultivars"

“On the other hand, the existence of morphological differentiation among cultivars from distinct geographical areas in the Near East and in the western Mediterranean region supports the second possibility in which wild local *sylvestris* germplasm significantly contributed to the generation of grape cultivars, possibly through multiple domestication events.”

“These results suggest the existence of at least two origins for grapevine cultivars, one in the Near East and a second one in the western Mediterranean region that gave rise to many of the Western European cultivars”

“Whether this second origin represents independent domestication events or developed as a consequence of the east to west transmission of the ‘wine culture’ will require further archaeological research.”

96. Although to determinate the existence of the bottleneck is a principal goal obtained by authors, I am surprised that results, graphs or values are omitted in the main text. Bottleneck in grapevine was observed in several papers. I look forward to seeing a discussion that also involves previously published articles on this topic.

Quantitative data for the effects of the domestication bottleneck are given in the first paragraph of the Result section using four metrics:

- (1) the degree of retention of ancestral (trans-specific) polymorphisms in *sativa***
- (2) the amount of variation present in *sylvestris* that had been lost in *sativa***
- (3) the nucleotide diversity in *sylvestris* and *sativa*.**

Additional details and the fixation rate in *sativa* of ancestral variation by age of the mutation are provided in numerical and graphical forms in Supplementary Figure 4.

All these metrics expressed and showed the genome-wide effects of the domestication bottleneck.

The global genetic effects of the domestication bottleneck are discussed in Lines 168-174 using selected papers (i.e. Myles et al 2011) that report homogenous and reliable comparisons with other quantitative estimations expressing the reduction in genetic diversity between wild and domesticated compartments. We omitted to cite and discuss estimations of the domestication bottleneck based on the reduction of effective population size (Zhou et al 2017 and Liang et al 2019), because the patterns of N_e are similar between wild and domesticated grapevines and the bottleneck revealed by N_e is more likely driven by global climate changes instead of human activities such as domestication, as explained by Liang et al 2019 in their paper.

Local effects of the domestication bottleneck are given by several metrics expressing local reduction of diversity in textual form in the paragraph “Genomic patterns of non-neutral evolution in cultivated genomes” and in graphical form in Supplementary Figure 34. The conclusions on the effects of the domestication bottleneck on local reduction of genetic diversity are extensively discussed in the same paragraph.

120. I don't understand why you show the Supplementary Figure 4. This analysis seems so far from the

aim of manuscript. Although your sampling could be adequate for a populations study, it seems inadequate to study the divergence times between species. However, too few information about how you have conducted the analysis are reported in the text.

Supplementary Figure 4 shows the effect of the domestication bottleneck and the loss of diversity in the domesticated compartment that the Reviewer has requested in the previous point.

It was also instrumental for identifying the mutation direction at each variant site, thus defining ancestral and derived allelic states.

145. Why have you choose $K=4$ respect to $K=2$? In the text is not explained and I am surprised that I have not found the likelihood values in the supplementary material. I believe that this choice is a fundamental step because it can strongly influence the results and consequently the discussion. I attend that you report more details and explain better your choice. Observing the little information reported, I would have chosen $K = 2$. Generally, it is better to avoid adding more groups unless their existence can be justified by a corresponding increase in the likelihood.

All options are now discussed in a thoroughly revised version of the paragraph “Population structure and ancestry components”, following the considerations explained above. Extra information has also been added to the revised version of Supplementary Information. Values for the variance of relatedness between populations explained by alternative models are now provided explicitly.

156-157. The authors said: “These two components could be attributed to the aberrans forms of *sylvestris* and deserve special attention to better understand the origin of wine grapes.” This sentence is not clear. I don't understand why you say that the two additional ancestral components (orange and gray) can be attributed to the aberrans forms of *sylvestris*. Can you explain how you infer this fact from the results?

Because the results indicate that, beyond the cultivated varieties, they are not detectable in Western and Eastern *sylvestris typica* and only found in Eastern feral grapes, which can be considered the closest group to the elusive form of the *sylvestris aberrans* wild progenitor.

However, in order to eliminate any source of confusion we renamed the four ancestry components as W1, W2 (those predominant in wild germplasm), C1, C2 (those mostly found in the cultivated compartment and for which tracing of the (extinct ?) wild donor population remains obscure, given the available *sylvestris* samples).

164: In figure 1B I don't find the “modern Western wine grape” group that you describe in the Text. What do you mean? Maybe that you mean a clade. If it is so, can you add this information in figure.

Figure 1B has been simplified as requested elsewhere.

165. You said: “...modern Western wine grapes deriving from introgression of western *sylvestris* individuals into table grapes”, I don't see in figure a migration from western *sylvestris* to table grapes. What am I missing? I see an intense migration on the tip of Alpine wine grapes and a weak migration on the common ancestor branch of the clade (Alpine wine grapes and Balkans and Magna Graecia).

Admixture between western *sylvestris* and table grapes generating wine grapes in the Alpine countries is predicted by residuals in the bifurcating tree (now reported in Supplementary Figure 13).

174. I could agree on the geographical division of the groups for *sylvestris* but you should insert in excel file the geographical origin of each accession (including geographic coordinate) that you have analyzed.

We have added the country of origin information in the Source Data.xls file. The geographic coordinates are included in the original papers and associated information that first reported SNP chip genotypic data for those accessions.

176. Why have you excluded varieties with highly admixed ancestry? In my opinion, these accessions should be the more interesting in a work that attend to explore ancient migrations between populations. I am surprise of this choice.

TreeMix requires clustering of genetic variation into discrete groups of well-differentiated individuals. Individuals with highly admixed ancestry blur these boundaries.

178. How you have assigned each group to a geographical area of origin or cultivation is not clear. You should explain better your choice. I underline that, for several varieties could be complex to assign a geographical area of origin but I remark that this choice is essential because forming groups with different criteria will lead to different results. Moreover, if I have understood correctly, so far you have divided several groups by geographical area. How do you justify the origin of other groups? for example, primitive European varieties group?

A *priori* geographic clustering was used only for producing for the TreeMix analysis shown in Figure 1B, with the rationale and the limitations explained in response to the general comments of the same reviewer. Individual varieties were assigned to a country of origin based on information reported in the Vitis International Variety Catalogue database or to the earlier or most renowned growing area. Countries were grouped into broad geographical areas that have homogeneous within-areas and differentiated between-areas climate conditions, following the Köppen-Geiger climate classification map.

The group identified as “Primitive varieties” was defined in lines 310-312 and its complete composition is reported in Source Data to avoid ambiguity. They fall short of other groups by their peculiarity of (1) having substantial fractions of all ancestry components, including an important fraction of the W1 wild Western component (Source Data) and (2) not appearing to be descendants of prolific progenitors (Supplementary Figure 30). As for their origin, this should be explained by the results of the TreeMix analysis and 3-population test.

204-208. I agree only partially with your sentence, (see between brackets the comments). You said: “TreeMix also suggested the occurrence of more ancient events of migration within the continental population of *sylvestris* (so far, it is clear), likely predating domestication and resulting from the Last Glacial period displacement (this affirmation is vague because you have not inferred divergence times on nodes), with evidence of southward and eastward migration from Northern Europe into the Mediterranean and Balkan populations (arrows from Northern Europe are multi-directionals. you should

to discuss the single migrations. I suggest to assign a letter to each migration and then discuss each migration) and from Southern Europe into Easternmost populations (migrations from glacial refugia are widely discussed in several papers for animals, plants and also grapes; these last migrations that you describe from Southern Europe into Easternmost populations are a novelty for me. I think that you should be more precise; perhaps you are confusing the effect of glaciation migrations with the spread of varieties through humans).” You should rewrite this sentence.

We removed this sentence from the main text. A short explanation of the predicted admixture events among populations of *sylvestris* shown in Figure 1b is given in Supplementary Figure 22 with pertinent literature references that have modelled the geographic retraction of temperate trees during the LGM.

209-2011. Also, I agree only partially with your sentence (see between brackets the comments): You said: “The scenario depicted by the population structure and relationship analysis presented here is fully consistent with the arrival of table grapes from the East to the Mediterranean Basin (I agree with you because the origin of table grapes is clear in Figure 1) followed by the occurrence of multiple hybridization events with different populations of local wild grapes (This second assertion conflict with you results; In figure 1B, I see only a flow between Table grapes and Eastern feral, but Mediterranean wild grapes are not involved; In figure 1C, I don’t see any migration).” You should rewrite this sentence.

This sentence has been changed, following the simplification of Figure 1b.

239-241. Can you deduce the origin of western wine grapes using a PCOA? I suggest to test your hypotheses using specific approaches such as Bayesian methods (they are proposed in several papers).

The PCA and PCoA (they are both provided in the main text and in Supplementary Information, respectively) only provided additional support in what we deduced from the inference models implemented in the software ADMIXTURE which are, indeed, based on Bayesian algorithms and by the predictive models of TreeMix.

Moreover, I have some questions about Figure 3. The interpretation of Figure is complex, and the symbols are too similar between them (including, also, the colors). The names of groups are different from Figure 1, this is an obstacle in the interpretation of grapevine origin. I suggest to use the same groups in both figures. Moreover: Why the accessions analyzed in this paper are only 4 (green square with cross)? What do you mean for admixed group? What do you mean for feral Adriatica and faux savage? These groups should be explained.

Figure 3 has been simplified. Group naming in panel 3a is the same as in with Figure 1a. Symbol colors for the 4 ancestral populations in the cultivated compartment are the same used for ADMIXTURE group assignment (Supplementary Figure 10). The symbols used are square (open and solid), triangle pointing up (open), cross and circle. They are the most differentiated ones, among those available in R. The text of the figure legend reports now a sentence for the symbols indicating sequenced accessions and accessions that were only genotyped with SNP chip in other papers. “Admixed” refers to the cultivated varieties in the WGS set that are not included into 4 ancestral populations (Supplementary Figure 10 and Source Data). “Feral adriatica” referred to 3 samples from the Adriatic

coast of Croatia which were indicated separately from other western European wild accessions in the previous graphical layout. In order to comply with the request above (i.e. highlighting the same grouping as in Figure 1), they are now included in the broader group of Western feral. The term “faux sauvage” is drawn from the articles where those samples were named as such. The use of this term now explained in the main text (Line 311).

244-248. I am surprised that you want to explain the origin of Lambrusco of Sorbara observing a PCOA that explain the 10% of variants. I think that you should use a different approach. Can you propose a test to verify the origin of Lambrusco? Moreover, I think that you should take into account the origin of family. Can you show where fall the other Lambrusco samples in PCOA? Instead, as you wrote, it just sounds like a speculation. However, the topic is interesting, I have found several papers focus that suggest the origins of Lambrusco family, placed between *sylvestris* and *sativa*. You should highlight what your results add to the previous studies.

It was stated in the original version of the manuscript that “The PCA gave us the opportunity to identify individual accessions that may serve as illustrations of the blurred boundaries between the wild and the cultivated compartments.” All examples, including Enantio and Lambrusco di Sorbara, were discussed in that paragraph in that specific context and with language that is not inducing the readers to mistake the PCA outputs for results of pedigree analysis.

As for the ancestry of the Enantio and Lambrusco di Sorbara genomes, in addition to the PCA, the readers are offered more robust information in support of the above statement from the ADMIXTURE analysis (Supplementary Figure 7), which shows a membership proportion for Enantio and Lambrusco di Sorbara fully compatible with carrying *sylvestris*–*sativa* hybrid genomes or deriving from early backcross generations.

We added this comment as well as literature references to both Lambruscos and Mansengs.

248-249. Hybridization events within the autochthonous *sylvestris* populations from Ketsch are well known. You should highlight how your results add information to the previous studies.

We added one reference referring to the case of the accession KE–06. In the new version, we also provide additional information on the amount of haplotype sharing with the most closely related cultivated varieties, which is not reported in any of the previous studies.

291-310. In these lines I read a miscellaneous of methods and results about the study of climatic variables. I suggest to split these informations in two distinct paragraphs. However, the results are not used to explain the diffusion of grapevine. Why have you applied these analyses?

To assess if there is a significant correlation at a continental geographic scale between individual ancestry components and climate conditions where individual genotypes have found their best growing conditions or where they are successfully grown.

Sentences about methodological aspects have been moved to the Mat&Met Section in the revised version.

I expect that you to discuss your results in a context of origin and diffusion of the grapevine. Other studies on this topic are proposed. I have added the title below; you should consider them to prepare a complete discussion.

Species distribution models are used for predicting the geographical range of species distribution based on current species distribution and environmental data. As explained in the original version of the manuscript, our aim is different. We sought to assess if genetic variation is correlated with climate parameters.

513-516. I don't understand why you have used two different proportional memberships to assign the admixed populations. If I understand correctly, you have used 0.85 as cut-off for cultivated varieties and 0.99 for wild plants. I believe that you should use the same method.

Because they reflect different types of population history in the wild and cultivated compartment.

Within each compartment, any value might be considered arbitrary and questionable. However, something between 0.8 and 0.85 is generally used in population genetic analysis in most crop species.

The threshold we imposed was much higher (0.99) outside of the cultivated compartment for filtering out, with a high degree of confidence, from the *bona fide* sample of *sylvestris* single individuals that may have resulted in Europe from crop-to-wild gene flow in small and geographically isolated wild populations.

We point out that those individuals were not discarded from the analyses. They were only grouped separately as feral grapes. Under the alternative hypothesis that those intermediate forms are not the result of crop-to-wild gene flow but rather represent paredomesticated remnants of an independent event of domestication in the West, the group of feral grapes should have shown closer relationships with the Western cultivated germplasm in other analyses (PCA and residuals in TreeMix), which was not the case shown by the data.

516-518. This choice is not clear: "For TreeMix analysis only, Eastern Feral grapes were considered as such only if they did not share first- or second-degree relationships with cultivated varieties."

The same as for the previous point. In order to filter out, with a high degree of confidence, single individuals that may have resulted in the East from recent hybridization events between wild and cultivated accessions, due to crop-to-wild gene flow, we applied stringent criteria for generating well differentiated groups of Eastern wild, Eastern feral and Eastern cultivated as an input for the TreeMix analysis.

518. Why have you used blocks of 300 SNPs? How have identify the appropriate block size? Have you applied a specific test to make this choice?

Please, see above the response to the same question in the General comments section.

520. I have not found the tables that report the ABBA-BABA results.

The results of the ABBA–BABA test are expressed by the f_d statistics that are plotted in Figure 2a (adjusted f_d values, individually in 2,368 genomic windows). We did not replicate the same information in a tabular format.

521. This choice is not clear: “ancestral = Table grapes”. Why have you used Table grapes as an ancestral group? Many of your assertions suggest that table grapes are of recent origin. I report only some your assertions: Line 50: Table grapes were also introduced eastwards into Central Asia along land trade routes. Line 165:introgression of western *sylvestris* individuals into table grapes. Line 198: ...wine grapes and table grapes have diverged for 2.6 K years. Why do you consider the table grapes as ancient? Wouldn't it be better to use *sylvestris*?

Our assertions state that table grapes had a pivotal role in origin of the other groups and TreeMix suggested that table grapes received introgression from *sylvestris* to generate Western groups of cultivated varieties. This was the hypothesis tested in the ABBA–BABA test.

The term “ancestral” identifies the population that received the introgression, which is ancestral with respect to the population that resulted from the introgression. *Sylvestris* is the population that we tested as the donor of the introgression. There is no need to make assumptions in the ABBA–BABA test on which group is more ancestral between the donor (*sylvestris*) and recipient (table grapes), because ancestral and derived states of each mutation are defined by the state in the fourth population (the outgroup).

In order to avoid any lexical ambiguity, the relative relationships are now called in the text (Lines 688-689): outgroup = *Vitis* sp., recipient = Table grapes, donor = Western *sylvestris*, generating = Alpine wine grapes.

522. What do you mean for “Alpine wine grapes”. I think that you mean *sylvestris*.

No, we mean “wine grape varieties from Alpine countries”. See above the response to same issue in the General comments section of the same Reviewer.

529. How have you produced the alignment by Muscle (including number of characters) and how have you filtered the alignment by Gblocks? These data are not reported.

By default parameters if not otherwise specified.

FIGURE 1: Number of samples on pie charts should be reported for each group. I don't understand the pie charts in the black box; sativa and *sylvestris*? Can you explain better?

Figure 1 has been simplified in the revised version. All extra information and explanations (group sizes, composition, proportional ancestry variation within groups and among groups) are provided in Supplementary Information.

My considerations are based on personal experience and the follow bibliography.

Geographic distribution of grapevine.
Laocucu et al., (2018) PLoS ONE 13(2): e0192540.
De Lorenzis et al. (2019) BMC Plant Biology, 19:7.
Arroyo-García, R. et al. (2006). Mol. Ecol. 15, 3707–3714.

Whole-genome scans using millions of variant sites may provide high power and resolution that do not necessarily confirm earlier assumptions and interpretations with fewer genotypic data and may disclose novel scenarios on a more robust basis.

Climate data
Araújo et al (2019). Science Advances 5: eaat4858.
Zurell et al., (2020). Ecography, 43: 1261-1277.
Garzon et al., (2019) New Phytol, 222: 1757-1765.

These citations refer to methods that were developed for modeling species' environmental requirements and are commonly used for mapping their possible range of future dispersal or cultivation based on their current distribution. This is not the scope and utilization the climatic data were used for in the submitted paper.

Lambrusco and Ketsch populations
Emanuelli, F. et al. (2013). BMC Plant Biol. 13, 39
Schneider et al., (2009) Acta Hort. 827
Grassi et al., (2003) Theoretical and Applied genetics. 107:1315-1320
Schroder et al., (2015) Can. J. Plant Sci. 95:1-8

We have taken these and other references into consideration for discussing the issues related to the Lambrusco varietal family and the Ketsch natural population.

Treemix
Pickrell, J. K. & Pritchard, J. K. (2012). PLoS Genet. 8, e1002967
Manual for TreeMix v1.1 Joseph K. Pickrell, Jonathan K. Pritchard October 1, 2012
Several papers in which the authors have used Treemix on plants.

The way we used TreeMix and handled strength and weaknesses is extensively described above.

Reviewer #2 (Remarks to the Author):

This ms by Magris et al. is a follow-up of the paper by Liang et al. (2019) on genome re-sequencing of 472 grapevine accessions. Liang et al. included most of the Vitis species and a number of Vitis vinifera accessions in their study and their work provided interesting data on Vitis phylogeny, on cultivated and wild V. vinifera population structure and demography, on identity-by-descent relationships among cultivars, on genes targeted by artificial selection during domestication. However, they acknowledged that their dataset did not include enough European grapevines (both wild and cultivated) to really explore the history of cultivated grapevines and the introgression with their wild counterparts in Europe.

Magris et al.'s study is an attempt to do this. Conceptually, both studies are quite similar, the major difference lies in the dataset. Magris et al. focused on the European accessions and they sequenced 124 new accessions (and analysed a total of 204 ones). They performed population structure and demographic analysis, cultivated grapevine pedigrees, GWAS, selective sweep analysis. These are the analyses that I found particularly interesting:

1- They studied introgressions among cultivated and wild European grapevines, and found that these introgressions have been pervasive, which explains why the European cultivars are different from other cultivars. They concluded that their data do not support a secondary domestication event in Europe (but see my comment below).

2- They analysed the distribution of the wild and cultivated grapevines and found correlations with some bioclimatic variables. A similar analysis was not in Liang et al. and provided some interesting data. Re-sequencing 124 accessions is less than 472 in Liang et al. but it is still a fairly impressive sequencing effort and large dataset. They provided important information on European grapevine cultivars, complementing Liang et al.'s paper. I think this ms merits a publication in Nature communications pending some revisions:

1) One key missing information is the geographic distribution of the 124 (and even 204) accessions studied here. There is an excel file in the Supplementary Material showing the list of studied accessions, but this list does not include the country of origin of the accessions. It is currently difficult to have an idea of how well the grapevine diversity (in particular the European one) is represented in the studied sample. This is very important as it can affect the conclusions. For example, the authors have tested the idea of a secondary domestication event in Europe. The secondary event was proposed to be located in the Iberian Peninsula, possibly in Portugal (their ref. 8). Sampling in this area is thus critical. Other analyses presented in the ms may be affected. The information about the country of origin should be added in the excel file.

This information has been added to the Source Data.xls file in a tabular format.

A figure with a map of Europe locating the studied accessions should also be provided. A table showing how much of the grapevine diversity of the main wine-producing European countries was studied here (simply computing the number of studied accession over the total number of accessions in a country) would also be very helpful.

The map showing the most representative site of cultivation of the studied accessions was provided in the original version in Figure 4 panels c and d. The figure Legend reported the statement "The most representative site of cultivation of each variety is indicated by a black dot."

We added a Supplementary Table (S3, with the new numbering) showing the estimated number of varieties (i.e. varieties that play a substantial role for wine production) by country. This estimation is based on Anderson and Nelgen (2020), who considered the list reported in the book *Wine Grapes* (2012) authored by Robinson, Harding and Vouillamoz as a detailed guide of 1368 commercially grown 'prime' varieties worldwide.

The list of varieties by country is now reported in the Source Data file.

2) I found the first parts of the Results and Discussion very interesting. I was less impressed by the last section (on non-neutral evolution), and found this section was a bit long. I think the ms could be

streamlined and written in a more straight-to-the-point way, the letter format may be even more suitable than the article one. In this case, Figures 5 and 6 could go to Supplementary material.

The last section reports on the characterization of regions of the genome that have undergone selective sweeps, which is one of the major goals of the work. We still think that this an integral part of the article and we think we have provided novel molecular evidence for a strong candidate gene underlying a major domestication related selective sweep. The suggestion to reduce the article to a letter format conflicts with the numerous requests from the other reviewers to add more and more detailed information and explanations.

3) P3, lines 107-114: it would be useful to add information on the quality of the data: % of mapped reads, estimated mapping error rates, genome coverage.

We added in Source Data the breakdown by accession of several metrics, among which:

- (1) the number of uniquely mapping reads (excluded those aligning to multiple locations in the reference and those there were flagged as library duplicates) and, using these reads,**
- (2) the mean read depth**
- (3) the fraction of the reference genome sequence covered**
- (4) the number of informative reference base positions**
- (5) the number of SNPs with respect to the reference genome sequence**

Estimates of genotyping error rates were given in the original Supplementary Information.

4) P3, lines 115-120: same as comment 3 for the SNPs. Please, give details on filtering and comment on the T_i/T_v rates.

The number of SNPs by variety with respect to the metrics above were also added to the Source Data file.

T_i/T_v is 1.57 in non-repetitive DNA regions of *sativa*, as reported in Supplementary Information. We also reported the breakdown of t_i/t_v values in exons (1.28), introns (1.27) and in the intergenic space (1.59). Although t_i/t_v is sometimes used as a quality indicator for the SNPs inferred from WGS data, t_i/t_v is not considered universal and, within each species genome, it is sensitive to variation among genomic fractions. Any estimate and comparison between literature reports suffer from biases due to the relative distribution of the SNPs considered in different genomic fractions.

5) P4, lines 130-132: "it still clearly shows that the domestication events that led to the creation of the cultivated varieties, unlike in other crops and fruit trees^{18,19}, did not lead to significant genome-wide losses of genetic diversity as a consequence of a major genetic bottleneck." Perhaps the authors could elaborate a little bit more and discuss and cite other papers, for example:

Qi J, Liu X, Shen D, Miao H, Xie B, Li X, Zeng P, Wang S, Shang Y, Gu X, Du Y, Li Y, Lin T, Yuan J, Yang X, Chen J, Chen H, Xiong X, Huang K, Fei Z, Mao L, Tian L, Städler T, Renner SS, Kamoun S, Lucas WJ, Zhang Z, Huang S. A genomic variation map provides insights into the genetic basis of cucumber domestication and diversity. *Nat Genet.* 2013 Dec;45(12):1510-5. doi: 10.1038/ng.2801. Epub 2013 Oct 20. PMID: 24141363.

Cubry P, Tranchant-Dubreuil C, Thuillet AC, Monat C, Ndjiondjop MN, Labadie K, Cruaud C, Engelen S,

Scarcelli N, Rhoné B, Burgarella C, Dupuy C, Larmande P, Wincker P, François O, Sabot F, Vigouroux Y. The Rise and Fall of African Rice Cultivation Revealed by Analysis of 246 New Genomes. *Curr Biol*. 2018 Jul 23;28(14):2274-2282.e6. doi: 10.1016/j.cub.2018.05.066. Epub 2018 Jul 5. PMID: 29983312.

We added the reference Qi et al, which reports comparisons of the domestication bottleneck effects in additional fruit crops, such as cucumber, watermelon and tomato. The second suggested reference refers to grain crops, which would open another field of discussion.

6) In different analyses, the generation time (GT) was set to 3 years. This seems a very small GT for grapevines. Wild grapevines are usually sexually mature after 5 years, and reach optimal reproduction after 7-10 years, some individuals can be >100 years old (see their ref. 10). The impact of GT's value on their conclusions needs to be discussed.

We used GT = 3 for two reasons. First, to generate and compare results that originated from the same methodology used in all other articles dealing with this sort of analyses in grapevine (Myles et al 2011, Liang et al 2019 and Zhou et al 2017). Second, we agree with the assumptions of those authors and to the others they had referenced to. Spontaneous seedlings—growing in the absence of human intervention that prunes canes back to basal buds to contain plant size in trellising systems—develop inflorescence primordia in an acropetal manner at distal nodes and complete the juvenile to adult phase transition in 3 years.

7) I found Supplementary figures 9 and 10 a bit hard to understand. I know that this kind of analysis can return complex networks. But here, most of the connections between the accessions are simply intractable. Simplifying the figure would help. Naming clusters and using different colours for different clusters would also help.

We offered the readers these figures in pdf format in order to make the network fully searchable for any PO-relationship that involves individual varieties, simply selecting Find under the Edit menu (or using the Ctrl+F shortcut) and typing the variety of interest.

Reviewer #3 (Remarks to the Author):

This submission by Magris and colleagues describes their impressive efforts to generate a massive amount of sequencing data on wild and cultivated *Vitis vinifera*. The data in the study will undoubtedly be used for by grapevine researchers for years, and it has relevance for evolutionary questions as well as for germplasm management. The authors' interpretations on grapevine domestication are also useful, although some of the findings have been made before, albeit on more limited datasets (e.g., SNP chips or whole genome sequencing of few individuals). Still, it is important that genome-wide data confirm previous understandings of a single domestication origin for cultivated grapevine and a limited genetic bottleneck.

I found it helpful to see this study as a counterpart to the Liang et al. (2019) paper, suggesting *Nature Communications* is a fitting journal for publication. The Liang et al. paper has more grape accessions, but their interest on different *Vitis* species means this paper has a different focus. Overall, I find Magris et al.'s paper to be of a high standard, but there are a few items that could be developed to improve the

study.

The most unexpected element of the study is the controlled garden experiments and related transcriptomics. From the abstract, it was unclear this is even a part of the project—it seemed it was only a genomic dataset. It would help to clarify this from the start, as it does give this project a different angle than that of Liang et al., Myles et al., Zhou et al., etc. In the main text, these experiments and the RNA data are mentioned in the very long section “Genomic patterns of non-neutral evolution in cultivated genomes”.

In the original version, it was stated at the end of the Introduction Section as the first bullet point objective (former Lines 96-98) that “characterize regions of the genome that have undergone selective sweeps” was one of the major goals of the work.

These experiments, including RNA-Seq and berry size variation analysis, were instrumental for supporting the role of two candidate genes in the sweep region on chromosome 17. These experiments were performed for that specific purpose. We think that they are described in the appropriate context within the Result section and they are necessary for the characterization of the domestication haplotypes in the sweep region. We have however now referred to the transcriptome analyses at the end of the abstract and at the end of the introduction section to explain what we did in order to characterize the sweep regions.

This section needs some further clarity and organization there is a single paragraph from lines 339-430. To heighten the reader’s comprehension, I suggest creating a new section, but at minimum implement some paragraphs and consolidate some of the discussion on genes and berry size. The alternative option would be to remove the RNA component and expand genomic analyses on some of the other areas of the genome with high levels of homozygosity.

Following the reviewer’s suggestion the section has been split in two parts, one dealing with the identification of the sweep and their candidate genes, the other describing the transcriptional and phenotypic effects of the domesticated haplotypes.

The section also ends on with the authors being “tempted to speculate”, which is an unusual tone for a scientific work. I recommend rephrasing into more a more empirical statement, or potentially writing as an open hypothesis to guide future research.

The tone of the sentence has been changed in the revised version.

One shortcoming of the study is the emphasis placed into the four ancestral groups. While there clearly is geographic structure in *V. vinifera*, the four groups are often treated like they are biological certainties. Indeed, the cross validation error plot shows a number of very similar values, and $K=3$ looks potentially better than $K=4$. Other researchers have used other values of K , like Bacilieri et al. (2013 BMC Plant Biology) using 3 main groups with an additional analysis of $K=5$ to evaluate structure within groups. At the least, the Admixture plots should be presented with several values of K for transparency sake (can be shown in supplemental document if needed). That is a common approach in the field, as exemplified by the Liang et al. study.

Admixture plots with different K values are included in the revised version of the Supplementary Information.

Please, see general and point-to-point responses to Reviewer 1 on the issue of the number of ancestral groups and the extensive revision of the related paragraph in the text.

Regarding the comparison with the cited literature reports on the best K values for discriminating between the main components of population structure and lower hierarchical levels, we want to point out that such comparisons are heterogeneous. Some analyses are based on a set of pre-ascertained variant sites (in the order of magnitude of K SNPs and pre-selected in a discovery panel). Other analyses interrogate millions of SNPs directly in the test panel, without any ascertainment bias. We have shown in Figure 3 the strong effect of the ascertainment bias on the separation provided by the PCA. Figure 3 shows, indeed, that a set of pre-ascertained K SNPs, while reproducing exactly the very same broad picture, may fail to capture a substantial part of the hidden variation that is otherwise disclosed by unbiased SNPs, allowing for a wider separation for two main components of the explained variance.

If we compared our K = 2 to 4 grouping with other studies based on WGS data, the results would be totally consistent with the conclusions of Liang et al. who considered from 2 to 4 ancestry components as reliable inferences in the species *Vitis vinifera* (red, green, brown, blue, in their Fig. 2; see also their Supplementary Figure 5).

Figure 1 needs to be improved in a number of ways. First, panels A and C should have more labels to allow readers to interpret samples/regions. Only Western *syvestris* and western feral are labeled, and the key in panel C is very difficult to see. I do not understand why panel A has two red areas and two black areas, apparently for wild and cultivated, respectively. These figures may need to be included in a larger format as supplemental figures will proper labels.

In the revised version, ADMIXTURE plots in panels A and C have been transferred to Supplementary Information, nested into the series of K plots showing incremental numbers of K ancestry components, as requested by two reviewers. Instead of using complex labelling, ADMIXTURE plots in are accompanied in their Figure legend in Supplementary Information by full information on the order of the accessions in the plot and on the varietal composition of each group.

In panel D, the TreeMix tree has some very strange features, like the Caucasian wine grapes being ancestral to the wild Caucasian grapes. This suggests something is wrong in the model, perhaps overfitting on a small number of datapoints or allowing too many migration events. At the least there should be more discussion of why this may not meet readers' expectations.

Yes, this was due to the inclusion of the group *syvestris* Turkey (*sy_TUR*), represented in the SNP chip public datasets by accessions with uncertain classification, which generated a number of migration events among groups of *syvestris*, including those that seemingly displaced the expected location of wild Caucasian grapes over the tree.

With the removal of that group, we obtained an increase in the explained variance of the relatedness between groups and we could highlight in a more simply and straightforward graphical visualization

the admixture between domesticated lineages and the wild populations than have generated primitive Western cultivars and wine grapes in the Alpine countries.

One of the goals of the project is stated to be associating genetic variation with climate. As far as I can tell, this was primarily done through by comparing environmental variables and latitude vs. the ancestry of four assumed ancestral groups.

vs. the ancestry components of each individual. Individuals are treated independently from ancestry groups in this analysis.

The problem is that this greatly depends on the ancestral groups being biological realities rather than the clusters of similar samples found in Admixture software. Presumably individuals within these clusters have different responses to some environmental conditions, and it would be quite intriguing to look at some key variables with a GWAS approach, as you did for berry ripening.

See responses to other reviewers and additional data in R1 on the reliability of the number of ancestry components.

Our approach was guided by the rationale that responses or adaptation to environmental conditions at a continental scale could be hardly detected by a GWAS approach, under the assumption that these traits are controlled according to an infinitesimal model. Indeed, a GWAS approach did not have enough statistical power to detect significant association with individual loci. We vice versa assumed that ancestry components may more broadly recapitulate variation in allelic frequencies at many loci and better capture cumulative effects at genome wide level.

Other points:

In the introduction you write “driven by human migration and maritime trades, paralleled by a transition in use from fresh consumption (table grapes) to winemaking (wine grapes)”. There is no citation for this, and some of the early archaeological evidence from Georgia indicates a very early use of grapes in winemaking (even though one may assume humans have been eating wild grape berries for millennia). Please update the question of table vs. winemaking based on the literature.

We have added a literature report addressing the issue (Migicovsky et al 2017). We replaced the term “transition” with “differentiation”, in line with Migicovsky et al 2017 and their results.

For Figure 3, the color pallet makes it difficult to differentiate many of the points. It would help to also vary the shape of the points in addition to the color.

Figure 3 has been simplified in the revised version.

Was any attempt made to look at the indels or transposable elements?

Yes, we used small Indel variation. With Indels, we explained a slightly lower % of the variance with the first two PCA components, compared to PCA with SNPs, but we obtained an identical distribution

of the explained variation in the bidimensional space. We did not find that result of particular interest and we excluded it from the original submission for the sake of concision. Following the reviewer's input, we have added it to as Supplementary Information in the revised version (Fig. S31c).

Did you correct for LD? This seems to be particularly important when cultivars have had limited sexual reproduction in the past centuries.

We considered the issue of LD and validated the results by the two approaches. A detailed description is given in response to the same issue raised by Reviewer 1.

In line 139, you don't specifically mention the analysis, just the authors. To clarify, suggest naming "TreeMix" here so that it is clear when the software is listed on line 163.

Yes, we added the name of the software that implements the cited method, when first mentioned.

Have any of the kinships identified in this study been at odds at previous genomic analyses? That could be an important insight in the importance of WGS vs. the efficiency of SNP chips.

Yes, we found some examples. The most relevant cases are represented by genuine full-sibling relationships that were previously mistaken for parent-offspring relationships and that remained unresolved with a limited number of variant sites included in SNP chips. We had not included that part in the original version because we felt it should have fallen short of the core of the paper. We have added now a couple of examples as Supplementary Information in the revised version (Fig. S25-S29).

Given that three areas of the genome have been identified as being under selection, this raises the question of how they feature in other analyses. Were the three loci on chromosomes 2, 15, and 17 included in the Admixture and TreeMix analyses? Their inclusion could cause cultivated accessions to appear more closely related than the rest of the genome indicates. It would be interesting to evaluate with both approaches.

Yes, in the sweep region on chromosome 2, the reduction of diversity is strong and the effects span over an interval of some million bp. This region was excluded from the original ADMIXTURE analysis. Thanks for having caught that point which was not mentioned in Mat&Met. Mat&Met is now updated. As for the other two sweep regions, one is strong but restricted to a short physical interval (chr17), the other spans a larger region (chr15), but it is weaker and covers a highly repetitive DNA region. We therefore found more appropriate to include variant sites across those two regions in those analyses.

I see the data have not yet been released for the main NCBI BioProject. Please ensure they are available at the time of publication.

Yes, they are deposited and the access is embargoed until article publication.

For Supplementary Figure 3, the three colors are not defined.

The shadowed area of the histogram represents overlap between the two distributions.

For Supplementary Figure 8, you should clarify N_e is the red line.

Yes, we added this information in the Supplementary Figure Legend.

Also, is the generation time of 3 years a realistic value? It surely works before vegetative propagation was commonplace, but I have questions on the reliability of this analysis for the most recent time when some varieties are maintained for centuries.

Please, see comments in response to the same issue raised by reviewer 2. Regarding the impact of vegetative propagation, we agree that the demography model makes full sense until approximately 900 years ago, when the start of vegetative propagation has been recently demonstrated (Ramos-Madrigal et al 2019). The most recent part of the predictive model is not discussed in the text.

For Supplementary Table 1, you have used commas for the decimal point, unlike the rest of the article.

Thanks for spotting it. Supplementary Table 1 (Supplementary Table 2 in the revised version) has been updated and commas were replaced.

Reviewers' Comments:

Reviewer #1:

Remarks to the Author:

I congratulate the authors who have improved the manuscript reporting numerous information that better explain their choices.

Only a few criticisms remain unsolved.

1- The trees of figure 1 should report the bootstrap values on each branch. Since you discuss also the nodes and gene flow that don't show the bootstrap, you should add the values everywhere.

2- How many trees with the same Maximum likelihood but different topologies have you found and how have you handle these trees is not clear. Have you produced a consensus tree? Can you explain better?

Moreover, I continue to be surprised that you want to explain the origin of Lambrusco of Sorbara using a PCOA

that explains about 10% of variants. You should propose a specific test to verify the hybridization origin. Moreover, you should extend the analysis to other Lambrusco varieties to verify if they have the same origin. You suggest that Supplementary Figure 7 support your hypothesis but in Figure I don't see the position of Lambrusco. Moreover, in figure S10 I don't find hybridization events for Lambrusco. Perhaps, You can use Treemix to identify the origin of Lambrusco, but I think that you need to have a sufficient number of accessions. Given this, if you have not specific results about the hybridization, you should avoid discussing in this manuscript the position of Lambrusco/Enantio.

Reviewer #2:

Remarks to the Author:

About my comment #1:

The sequenced accessions include very few iberian accessions (information now available in the data source file). 6 Spanish and 1 Portuguese accessions, while these countries have 162 and 253 prime cultivars respectively (information now available in table s4). I think they should tone down their claim about the rejecting the possibility of a second domestication center in the iberic peninsula. They should include more iberian accessions to really test this hypothesis.

More generally, it should be acknowledged that their set of sequenced European accessions is not representative of the diversity of the European grapevines. It is strongly biased towards German and Italian accessions, which make 63 and 48 out of the 204 sequenced *V. vinifera* accessions used in this study.

My other comments have been addressed.

Reviewer #3:

Remarks to the Author:

The authors' revised manuscript is a substantial improvement on the original submission, and the core results for the genome-wide analyses for cultivated grapes are deemed more reliable with the expanded testing. I'm impressed with the level of attention paid to the critiques from the three reviewers, as each comment was discussed and subsequently implemented or rejected for a specific reason. The text has been revised in many cases based upon the comments, and more importantly, several analyses were updated, such as correcting for LD and additional runs of TreeMix. Reviewer 1 had the most detailed discussion on the primary analyses, and perhaps some weaknesses will be spotted. That said, the results are internally consistent and generally support past research. Ultimately other bioinformaticians will download this large dataset and analyze it with other software, so there are ample opportunities for further scrutiny and discovery. All in all, I am pleased with the revision of the manuscript and support publication.

Point-by-point response to the reviewers' comments

Reviewer #1 (Remarks to the Author):

I congratulate the authors who have improved the manuscript reporting numerous information that better explain their choices.

Thanks for your appreciation.

Only a few criticisms remain unsolved.

1- The trees of figure 1 should report the bootstrap values on each branch. Since you discuss also the nodes and gene flow that don't show the bootstrap, you should add the values everywhere.

Bootstrapping is commonly used in tree-like based graphical representation of sequence-based phylogeny in the absence of interspecific hybridization and reticulation.

On the contrary, the vast majority of literature reports, ranging from papers published by the original developers of TreeMix (e.g. Inference of Population Splits and Mixtures from Genome-Wide Allele Frequency Data, Plos Genetics 2012) to the most recent ones (e.g. Haplotype-resolved genome assembly provides insights into evolutionary history of the tea plant *Camellia sinensis*, Nature Genetics 2021) do not use bootstrapping for supporting the graphical outputs of TreeMix analysis.

TreeMix analysis is chiefly aimed at detecting introgression signatures and is based on the statistics of the residual fit between groups, which we fully provided as Supplementary Information, and is independent from the topological position of each group in the tree-like based graphical representation. For the sake of reviewer's curiosity, we have added bootstrap values in the R1 version of Fig.1 and followed the common practice used in tree-like species phylogeny of showing only those values that surpass a certain arbitrary threshold, which we set at 70/100.

Discussions in the text on relationships and gene flow between groups are based on multiple lines of evidence that are provided not only by the TreeMix output but also by other analyses that gave consistent results from independent methods (e.g. haplotype sharing and IBD-based distance parameters, ADMIXTURE, 3-population test).

2- How many trees with the same Maximum likelihood but different topologies have you found and how have you handle these trees is not clear. Have you produced a consensus tree? Can you explain better?

See the legend of Fig.1 and the response to the previous comment. As for the graphical representation of the TreeMix output, we presented a random tree, 100 iterations were performed and bootstrap values indicate how many trees support the topology shown there.

Moreover, I continue to be surprised that you want to explain the origin of Lambrusco of Sorbara using a PCOA that explains about 10% of variants.

In the R1 version, in support of the PCA that was used, among other aims, to illustrate the intermediate position of these accessions between Western *sylvestris* and other cultivated varieties, we included data from ADMIXTURE analysis showing that the diploid genomes of Enantio and Lambrusco di Sorbara carry approximately 50 % of the W1 ancestry component. The W1 ancestry component makes up 100 % of the genome in *sylvestris* accessions from Western Europe.

We believe that this was a strong evidence in support of our statement on the origin of Enantio and Lambrusco di Sorbara. We furthermore stress that the fact that the first two coordinates of the PCA account for 10% of the variation (and not variants) is by no means a proof of the fact that the population structure identified is not robust. These percentages need to be interpreted relative to the size of the data matrix (and our SNP matrix is very large) because large datasets can capture a small percentage and yet still be effective.

You should propose a specific test to verify the hybridization origin.

In the R2 version, we have provided additional evidence derived from another test based on haplotype sharing analysis.

This test provides detail into the ploidy of the wild ancestry contribution by showing that the level of haplotype sharing (expressed as 1-haplotype distance) between Enantio and Lambrusco di Sorbara, on one side, and Western *sylvestris* accessions, on the other side, is exactly intermediate when compared to the level of haplotype sharing among Western *sylvestris* accessions and the level of haplotype sharing between *orientalis* table grapes (which are expected to have received no gene flow from Western *sylvestris*) and Western *sylvestris* accessions. We also showed that at least one haplotype in nearly all genomic windows of Enantio and Lambrusco di Sorbara is shared with at least one accession of the population of Western *sylvestris*.

This evidence is provided as Supplementary Figure S33.

We believe that all these lines of evidence combined leave no room for doubt as to the origin of Enantio and Lambrusco di Sorbara.

Moreover, you should extend the analysis to other Lambrusco varieties to verify if they have the same origin.

The results of ADMIXTURE analysis from SNP chip data were available for other varieties of the Lambrusco group in the "Source_data.xls" file, spreadsheet "diversity panel n=1445 profiles" that accompanied the R1 submission.

The contribution of the Western *sylvestris* ancestry component in Lambrusco Maestri, Lambrusco Salamino, Lambrusco Marani, Lambrusco Casetta is similar to that found in Enantio and Lambrusco di Sorbara. The Western *sylvestris* component was approximately halved in Lambrusco Grasparrassa.

Varieties of the Lambrusco group are not necessarily the result of independent hybridization events with *sylvestris* accessions and can be the result of backcrosses of *sativa-sylvestris* hybrids to other cultivated varieties, thus progressively diluting *sylvestris* introgression and the membership proportion in ADMIXTURE analysis.

All of this information is readily accessible to the readers who are interested in it, but we don't think this level of detail is interesting to most of the potential readers to deserve further explanation in the main text.

You suggest that Supplementary Figure 7 support your hypothesis but in Figure I don't see the position of Lambrusco.

The position of Enantio and Lambrusco di Sorbara was indicated in the Supplementary Figure 7 legend of the R1 version. The tabular version of the ADMIXTURE data in support of our hypothesis was provided as "Source_data.xls" file, spreadsheet "WGS Ancestry V. vinifera" that accompanied the R1 version.

Moreover, in figure S10 I don't find hybridization events for Lambrusco. Perhaps, You can use Treemix to identify the origin of Lambrusco, but I think that you need to have a sufficient number of

accessions. Given this, if you have not specific results about the hybridization, you should avoid discussing in this manuscript the position of Lambrusco/Enantio.

We think that the ADMIXTURE analysis data combined with the PCA analysis and the new evidence from haplotype sharing analysis provided in R2 are strong arguments for supporting the discussion on the *sativa-sylvestris* hybrid nature of Enantio and Lambrusco di Sorbara.

Reviewer #2 (Remarks to the Author):

About my comment #1:

The sequenced accessions include very few iberian accessions (information now available in the data source file). 6 Spanish and 1 Portuguese accessions, while these countries have 162 and 253 prime cultivars respectively (information now available in table s4). I think they should tone down their claim about the rejecting the possibility of a second domestication center in the iberic peninsula. They should include more iberian accessions to really test this hypothesis.

More generally, it should be acknowledged that their set of sequenced European accessions is not representative of the diversity of the European grapevines. It is strongly biased towards German and Italian accessions, which make 63 and 48 out of the 204 sequenced *V. vinifera* accessions used in this study.

Regarding the issue of the representativeness of genetic diversity of the Iberian sample in the WGS panel, it should be taken into account that:

- 1. The seven varieties with predominant cultivation in the Iberian peninsula that were included in the WGS panel (Airen, Tempranillo, Garnacha, Carignan, Listan Negro, Graciano, Touriga Nacional) account for 52.5 % of total vineyard surface area in Spain and 16.4 % in Portugal, as of 2010.**
- 2. Most importantly, genetic diversity analysis in the SNP-chip diversity panel, which included 182 cultivated varieties with predominant or exclusive cultivation in the Iberian peninsula showed that the set of 123 cultivated accessions used for WGS-based analyses covered well the PCA-space of the entire genetic diversity of the Iberian germplasm. We have added a new Supplementary Figure that illustrates this concept.**
- 3. The genealogical network in Supplementary Figure 30 shows that most of the minor varieties that are cultivated exclusively in the Iberian peninsula are close relatives to one another**
- 4. In population genetic analysis within cultivated compartments, which depart significantly from random mating populations, the relative number of individuals should be approximately proportional to the breadth of genetic diversity captured by each group of varieties. Departing from these principles and including larger numbers of highly related (by descent) individuals is not only uninformative but also detrimental, leading to allele frequency inflation and generating a bias for those analyses that rely on allele frequency estimations.**

The PCA shown in Fig. 3 of the main text provides strong arguments for stating that the WGS panel is representative of the currently known grapevine diversity, which was previously subject to SNP chip analyses by the curators of the largest European germplasm repositories, regardless of the absolute number of varieties representative of each country, as exemplified by the case of the Iberian peninsula shown in Fig. S32.

Regarding the issue as to the possibility of a secondary domestication center in the Iberian peninsula

- 1. The analyses we performed seem to clearly indicate that:**

- a. **Iberian cultivated varieties, as a whole, do not stand closer than other cultivated varieties to the PCA-space populated by Western European *sylvestris*, which would be otherwise expected if they had originated from a secondary domestication event in the Iberian peninsula or in the West**
 - b. **Iberian cultivated varieties show a predominant ancestry component (C1) that is absent from Western Europe *sylvestris***
 - c. **Those Iberian cultivated varieties that show a minor Western Europe *sylvestris* ancestry component (W1) are close descendants of Savagnin Blanc (Supplementary Figure 30) and stand midway in the PCA-space between Savagnin Blanc and table grapes (see PCA in Fig. 3 and the same PCA in Supplementary Figure 32 where the positions corresponding to Iberian varieties have been highlighted)**
2. **While our manuscript was under review, a resequencing study that is focused on some 40 Iberian cultivated varieties has been deposited on March 3, 2021, in the preprint server bioRxiv.org by Freitas and coworkers, who showed evidence for signatures of post-domestication *sylvestris* hybridization and introgression in Iberian cultivated varieties in the context of a single domestication event in the species and no sign of local neodomestication**
 3. **While Freitas and coworkers' conclusion is fully consistent with our results and interpretation, we could not integrate this raw data in our analyses because the resequencing study of Freitas and coworkers is based on low sequencing coverage and most of the accessions have 3 to 5 median read depth (Table S1, <https://www.biorxiv.org/content/10.1101/2021.03.03.432021v1.full>), which would cause a substantial fraction of heterozygous sites to be called homozygous due to random sampling**

We cannot exclude the occurrence of rare cases of *sativa-sylvestris* hybridization also in the Iberian peninsula, similar to those we reported for Lambrusco di Sorbara and Enantio, that may have occasionally generated hybrid accessions directly brought in cultivation. However, our data—now corroborated by an independent although not yet peer-reviewed resequencing study—make us confident to rule out the possible occurrence of secondary domestication events that substantially contributed to the germplasm currently used for cultivation in the West, including the Iberian peninsula.

My other comments have been addressed.

Reviewer #3 (Remarks to the Author):

The authors' revised manuscript is a substantial improvement on the original submission, and the core results for the genome-wide analyses for cultivated grapes are deemed more reliable with the expanded testing. I'm impressed with the level of attention paid to the critiques from the three reviewers, as each comment was discussed and subsequently implemented or rejected for a specific reason. The text has been revised in many cases based upon the comments, and more importantly, several analyses were updated, such as correcting for LD and additional runs of TreeMix. Reviewer 1 had the most detailed discussion on the primary analyses, and perhaps some weaknesses will be spotted. That said, the results are internally consistent and generally support past research. Ultimately other bioinformaticians will download this large dataset and analyze it with other software, so there are ample opportunities for further scrutiny and discovery. All in all, I am pleased with the revision of the manuscript and support publication.

Reviewers' Comments:

Reviewer #1:

Remarks to the Author:

1- I don't understand this disdain in showing bootstrap values. Even if some papers do not report the values, I believe that the authors, also for greater clarity, should show all values. Treemix, unlike other software, was developed to show both phylogeny and introgression. Moreover, bootstraps were not meant to respond to reviewers' curiosity, and, unlike what the authors say, the common practice used in species phylogeny is to show the bootstrap values on all nodes; especially on the nodes that you discuss.

2- If I understand correctly, you have found only one topology with the highest likelihood. I had the doubt because, when I have tried to use treemix, I found different topologies with the highest likelihood. Probably the large amount of data, that you have analysed, overcame this problem.

Reviewer #2:

Remarks to the Author:

please cite Freitas et al. indicating that their results on a larger set of Iberian accessions (including wild accessions) are consistent with yours obtained with a smaller set.

Reviewer #4:

Remarks to the Author:

Based on the Editor's suggestion, this reviewer only checked the soundness of the association mapping/GWAS parts (specifically results presented in Figure 4, Figure S43, and Table S4-6). Based on the limited information provided, there is no obvious flag. However, this reviewer cannot find what exactly is the statistical model used to conduct association mapping that leads to Fig S43. For instance, how do you control Type-I error and population structure in the association mapping? Could you please present the QQ plot to show that?

REVIEWER COMMENTS

Reviewer #1 (Remarks to the Author):

1- I don't understand this disdain in showing bootstrap values. Even if some papers do not report the values, I believe that the authors, also for greater clarity, should show all values. Treemix, unlike other software, was developed to show both phylogeny and introgression. Moreover, bootstraps were not meant to respond to reviewers' curiosity, and, unlike what the authors say, the common practice used in species phylogeny is to show the bootstrap values on all nodes; especially on the nodes that you discuss.

The reasons for the bootstrap values being dispensable were explained in the rebuttal letter that accompanied the previous submission. The TreeMix topologies shown in Figure 1 explain 99.8 % and 99.2 % of the variance in relatedness among groups in panels *a* and *b*, respectively. The statements in the manuscript on the relatedness among groups not only are based on TreeMix nodes that collectively explained >99% of variance but also are supported by concordant evidence that was obtained from independent analyses (3-pop test, haplotype sharing analysis, matrix of pairwise genetic distances).

2- If I understand correctly, you have found only one topology with the highest likelihood. I had the doubt because, when I have tried to use treemix, I found different topologies with the highest likelihood. Probably the large amount of data, that you have analysed, overcame this problem.

Yes, it is the most likely explanation for the difference you observed.

Reviewer #2 (Remarks to the Author):

please cite Freitas et al. indicating that their results on a larger set of Iberian accessions (including wild accessions) are consistent with yours obtained with a smaller set.

Yes, we added the suggested reference and an explanatory sentence in the text.

Reviewer #4 (Remarks to the Author):

Based on the Editor's suggestion, this reviewer only checked the soundness of the association mapping/GWAS parts (specifically results presented in Figure 4, Figure S43, and Table S4-6). Based on the limited information provided, there is no obvious flag. However, this reviewer cannot find what exactly is the statistical model used to conduct association mapping that leads to Fig S43.

The details of the GWAS models used for this analysis are now added to the Mat&Met Section of the main text. We apologize for this inconvenience. This paragraph went lost when transferring information from the main text to Supplementary Methods and *vice versa* across different versions of the manuscripts.

For instance, how do you control Type-I error and population structure in the association mapping?

This information is now included in the newly added paragraph of Mat&Met. Briefly, population structure was accounted for by using the option of generating genomic control adjusted p values. Type-I errors we controlled using a max(T) permutation test.

Could you please present the QQ plot to show that?

Yes, we added the Q-Q plots of unadjusted and GC p -values that are now shown in a newly added panel of Supplementary Figure S43.

Reviewers' Comments:

Reviewer #1:

Remarks to the Author:

I have not other comments

Reviewer #4:

Remarks to the Author:

The authors have assessed my comments nicely. No other concerns.